# An Overview of the Firefly Genus *Pygoluciola* Wittmer, a Phylogeny of the Luciolinae Using Mitochondrial Genomes, a Description of Six New Species, and an Assessment of a Copulation Clamp in This Genus (Coleoptera: Lampyridae: Luciolinae) [note 1]

**DOI:** 10.3390/insects16040394

**Published:** 2025-04-08

**Authors:** Xinhua Fu, Lesley Ballantyne

**Affiliations:** 1College of Plant Science and Technology, Huazhong Agricultural University, Wuhan 430070, China; fireflyfxh@mail.hzau.edu.cn; 2Hubei Insect Resources Utilization and Sustainable Pest Management Key Laboratory, Wuhan 430070, China; 3Firefly Conservation Research Centre, Wuhan 430070, China; 4School of Agricultural, Environmental and Veterinary Sciences, Charles Sturt University, P.O. Box 588, Wagga Wagga, NSW 2678, Australia

**Keywords:** firefly, mitochondrial genome, phylogenetic analysis, morphology, female reproductive system, integrative taxonomy

## Abstract

The firefly genus *Pygoluciola* was thought to be rare, as few specimens were to be found in museums. It is now known from 28 species, many from mainland China. While in the past, we used features of the males to identify genera and species, now we can increasingly use features of females, as well, as we associate females with 17 of these species.

## 1. Introduction

The genus *Pygoluciola*, first described by Wittmer (1939) for a single species, *P. stylifer*, was distinguished by pronounced male terminal abdomen modifications, to which it owes its name [1]. The posterior median areas of both abdominal ventrite 7 and abdominal tergite 8 are narrowly prolonged, with the apex of ventrite 7 being upturned. In dried pinned specimens, this area is partly covered by the downturned apex of tergite 8. These modifications are now known to occur in several other species [2].

Initially perceived as rare due to limited museum specimens [2], we now recognise 28 species from India, Sri Lanka, China, Malaysia, Indonesia, the Philippines, and northern Australia [2]. Some of that perceived rarity can be attributed to the occurrence of many species as solitary fliers [3,4,5,6,7]. Notably, the mass synchronously flashing displays of *Pygoluciola qingyu* Fu & Ballantyne in China have become a tourist attraction [8,9].

This genus is also unusual among the Luciolinae, which has a male focussed taxonomy, having data on females of 17 species and larvae of 5 [2].

The taxonomic status has changed, with *Pygoluciola* fluctuating between being a genus [1] and subgenus of *Luciola* Laporte [10], now reestablished as a genus; see [2,11] for an overview, with a clear affinity to a newly described genus *Abscondita* Ballantyne [12].

In addition to the abdominal modifications, *Pygoluciola* has been characterised by a suite of male genitalic modifications [2]. Recent discoveries include male forms lacking traditional terminal abdomen modifications yet sharing these distinctive genitalic traits [2,8,11]. Currently, five morphological groups exist, unified by shared aedeagal sheath and aedeagus features [11], but differing in the outlines of the male terminal abdomen.

However, genetic analyses remain limited, usually focussing on no more than two species of *Pygoluciola*, *P. qingyu*, and an unidentified species, and none has used representatives of the original morphological form consistent with *P. stylifer* [13,14,15]. Unidentified species are not illustrated or described, so their morphology is unknown, and morphological verification of either genus or species is not possible. Only in [16] are both species identified, including *P. qingyu* (without obvious terminal abdomen modifications) and *P. dunguna* Nada (with a modification of the terminal abdominal ventrite). No analysis presents an extensive phylogeny using more than 11 of the known Luciolinae genera [14,17,18,19]. Most show a relationship of *Pygoluciola* with species of *Abscondita* [13,14,16,17,18].

The possibility of a copulation clamp in *Pygoluciola kinabalua* Ballantyne was suggested [20], based on unusual female abdomen modifications. These include the swelling and hardening of the dorsal surface of tergite 7, a transverse ridge in the median area of ventrite 7 anterior to the deeply emarginated posterior margin, and the strongly hooked anterolateral extensions of its anterior margin [20] (p. 373). A copulation clamp is a system of male terminalia that engage the female abdomen during mating, and it has only been conclusively demonstrated within the Luciolinae in two species of *Pteroptyx* [2].

This study, a collaboration between a taxonomic entomologist and a molecular biologist, helps overcome existing identification challenges, advances *Pygoluciola* taxonomy, and clarifies the phylogenetic relationship of *Pygoluciola* with other Luciolinae. This is achieved in six steps: 1. *Pygoluciola* is redescribed; the five current morphological subgroups are evaluated, providing revised definitions and diagnoses for species within each group; 2. Six new *Pygoluciola* species are described using features of males, and also females (three species) and larvae (two species); 3. *Luciola davidis* Olivier is redescribed, assigned to *Pygoluciola*, and distinguished from *P. qingyu*; 4. Keys to all 28 currently recognised *Pygoluciola* species are presented, for both males and females (where available); 5. The possibility that the male terminal abdomen modifications in some species might function as a copulation clamp is investigated not from males but the female abdomen of *P. kinabalua*; 6. The mitogenomes of seven *Pygoluciola* species from two different morphological groups are sequenced and integrated in a Luciolinae phylogeny.

## 2. Materials and Methods

### 2.1. Morphology and Taxonomic Characters

Avoiding excessive self citation in this journal complicates the accurate presentation of taxonomic information for L.B., a firefly taxonomist. Conventional taxonomic tables will be omitted, but referencing recent papers should mitigate most issues.

Characters of males, females, and larvae used and methods of dissection are outlined [2,11].

We redefine key morphological features for clarity and reevaluate and build on Wijekoon’s classification [11] of *Pygoluciola* species into five morphological groups, providing revised definitions and diagnoses for species within each group.

Body length is taken as median maximum length pronotum plus length of elytron, to overcome the tendency of the pronotum in pinned specimens to droop. The head is not included in this measurement, as it may be variously retracted within the prothoracic cavity.

Elytral interstitial lines are numbered from 1 nearest the suture to 4 near the lateral margin; the degree of elevation is measured relative to the elevation of the sutural ridge; well defined lines are comparable in elevation and width to the suture.

All legs of *P. guigliae* and *P. stylifer* are now interpreted as curved, following a reassessment by Ballantyne of the actual type.

Abdominal segmentation: The ventral plate is termed a ventrite, and the numbering differs between males and females, as males have no obvious or visible ventrite 8 (see Table 1).

Aedeagi are measured horizontally with the median lobe uppermost. Length is taken from the anterior margin of the basal piece to the tips of the longest lateral lobe, while width is measured across the widest portion. Length of lateral lobes is taken from their base at the side, just below the posterior margin of the basal piece to the tip. Median lobe length is from base to apex.

Aedeagal measurements are classified as follows: Aedeagus short broad (length/width 2–2.5) [2] (figure 359); aedeagus elongate, longer than wide (length/width 2.7–3.5); lateral lobes short (length lateral lobes/length aedeagus 0.2–0.3); lateral lobes long (length lateral lobes/length aedeagus 0.4–0.6); median lobe short (length median lobe/length aedeagus 0.5–0.6) [2] (figure 370); median lobe long (length median lobe/length aedeagus 0.7–0.9).

Nomenclature: we use the English name for the type locality, which is anglicised and latinised in all new species, except for *quzhou* and *baise*, as the specific names represent the cities for which the species are named.

Photography: Ballantyne used an Olympus SC100 camera mounted on an Olympus SZX12 stereo microscope. Fu used a DP72 CCD camera mounted on a Nikon SZX 16 stereo microscope.

SEM: For detailed observations of the larval head of *P. tunchangia* and *P. manmaia*, we used scanning electron microscopy (SEM). Larval heads were cut off and then rinsed twice in phosphate buffer at 10 min intervals and post-fixed for 3-4 hrs in 1% osmium tetroxide at room temperature, followed by two rinses in phosphate buffer and dehydration in a graded series of ethanol, with 12 h stays at each concentration (30, 50, 70, 80, 90, and 100%). The specimens were then placed in acetone for two 12 h periods before finally being subjected to critical point drying. Each larva was attached with double-sided sticky tape to an aluminium stub and sputter-coated with gold to a thickness of about 200 nm. The specimens were observed under a JSM-6390 LV scanning electron microscope at an accelerating voltage of 25 kV [21].

Flash pattern analysis: Field conditions of temperature and humidity, as well as flash activity in relation to time after sunset, were recorded. In the field, flash patterns of *P. yupingia* were recorded by a digital video camera (Sony Digital Zoom 2000X, Tokyo, Japan) equipped with a photo intensifier λ-300EX (KnewTrino Corp., Tokyo, Japan). Recordings were later converted to the .avi file format, and we used software developed specifically for the analysis of these .avi format recordings (output of flash signals rendered as relative intensities against time). The flash duration, interval, and rate were calculated [22].

Dissections of bursa plates were performed by different people on different continents, and Ballantyne did not see all of these final female dissections. Because of the potential of subjectivity in the interpretation of the presence or absence of these plates, we have amplified our descriptions where none was seen to indicate also that “absence not assumed”.

To avoid confusion, figure references attached to the literature are written as “figure”, while references to figures within this manuscript are written as “Figure”.

Types are lodged in the Natural History Museum Huazhong Agricultural University, Wuhan.

Abbreviations for taxonomic characters and depositories of specimens are listed at the end of this paper.

### 2.2. Sample Collection for Molecular Analysis

Specimens collected from different localities were deposited in NHMHAU. For genome sequencing, specimens Fu collected were immediately preserved in anhydrous ethanol, followed by preservation at −40 °C in the laboratory prior to DNA extraction. A single individual was used to extract DNA.

### 2.3. DNA Extraction Sequencing, Assembly, and Annotation of Mitogenomes

Total DNA was extracted from the whole insect body using a modified CTAB method [23]. RNase A was used to remove RNA contaminants. The integrity and concentration of the DNA were verified by 1% agarose gel electrophoresis and Qubit fluorometry. The electrophoresis result showed a clean single-band product, and the DNA concentration was higher than 100 ng/µL. Then, 300 bp short-insert libraries were prepared following Illumina’s instructions. A BGI MGISEQ-2000 platform was then employed for whole-genome sequencing according to the standard protocols. The adaptors were removed from the sequencing reads using Fastp (version 0.24.0) and checked with FastQC (version 0.11.8). Clean reads were used to produce assembly using Geneious prime (version 2024.0.5) (GraphPad Software LLC d.b.a Geneious) map to reference *Py. qingyu* mitogenomes (MN688374.1). Genomic annotations were performed using Mitos2 (version 2.1.9) [24] and tRNAscan-SE (Version 2.0.6) [25]. The mitogenome sequences of 7 species of fireflies, namely, *Pygoluciola tunchangia*, *Pygoluciola yingjiangia*, *Pygoluciola baise*, *Pygoluciola manmaia*, *Pygoluciola yupingia*, *Pygoluciola davidis*, *Pygoluciola quzhou* were obtained.

### 2.4. Phylogenetic Analysis

Mitogenome sequences of the following were downloaded from the NCBI’s nucleotide database (GenBank): *Abscondita anceyi* (Olivier) (MH020192), *Abscondita cerata* (Olivier) (MW751423), *Abscondita terminalis* (Olivier) (MK292092), *Aquatica lateralis Japan* (LC306678), *Aquatica leii* (Fu et Ballantyne) (KF667531), *Aquatica wuhana* Fu et Ballantyne (KX758086), *Aquatica qingshen* Fu et Ballantyne (OM135505.1), *Aquatica xianning* Fu et Ballantyne (OM135504.1), *Asymmetricata circumdata* (Motschulsky) (KX229747), *Curtos bilineatus* Pic (MK292114), *Curtos costipennis* (Gorham) (MK609965), *Curtos fulvocapitalis* Jeng Yang et al. (MW582616), *Nipponoluciola cruciata* (Motschulsky) (AB849456), *Luciola curtithorax* Pic (MG770613), *Pteroptyx maipo* Ballantyne (MF686051), *Pyrocoelia rufa* Olivier (AF452048), *Sclerotia fui* Ballantyne (OL944083), *Tribolium castaneum* (NC_002081), and *Pygoluciola qingyu* Fu et Ballantyne (MN688374), as well as the mitochondrial genome sequences of 3 unknown species belonging to the *Pygoluciola* genus: *Pygoluciola* sp. (MZ571356), *Pygoluciola* sp. FM18 (MK292102) (collected from Mengla, Yunnan Province, China) [14], and *Pygoluciola* sp. (OP747324) (collected from Thailand).

PhyloSuite v1.2.2 [26] was used to extract 13 PCGs, 22 tRNA genes, and 2 rRNA genes. A total of 37 sequences were aligned in batches with MAFFT [27] using ‘-auto’ strategy and normal alignment mode. The alignments were refined using the codon-aware program MACSE v. 2.03 [28], which preserves the reading frame and allows incorporation of sequencing errors or sequences with frameshifts. Gap sites were removed with trimAl [29] using the “-automated1” command. ModelFinder [30] was used to select the best-fit model using the BIC criterion. Maximum likelihood phylogenies were inferred using IQ-TREE [31] under the GTR + R4 + F model for 20,000 ultrafast [32] bootstraps, as well as the Shimodaira–Hasegawa–like approximate likelihood-ratio test [33]. MrBayes 3.2.6 [34] was used to perform a BI analysis. The analyses of each dataset were performed with 4 MCMC chains and run for 20 million generations. Every 1000th generation was sampled as a consensus tree. The type of consensus tree was Halfcompat. The convergence of the independent runs was indicated by a standard deviation of split frequencies <0.01 and an estimated sample size (ESS) > 200. The initial 25% of sampled data were discarded as burn-in data, and the remaining trees were used to represent the values of posterior probability (PP).

### 2.5. Data Availability

The newly sequenced mitogenomes were submitted to the GenBank database under the accession numbers of *Pygoluciola baise* sp. nov. (PV069996), *Pygoluciola manmaia* sp. nov. (PV069997), *Pygoluciola quzhou* sp. nov. (PV081576), *Pygoluciola tunchangia* sp. nov. (OM201323), *Pygoluciola yingjiangia* sp. nov. (OM201324), *Pygoluciola yupingia* sp. nov. (PV069998), and *Pygoluciola davidis* comb. nov. (PV069999).

## 3. Results

### 3.1. Outcomes

The genus *Pygoluciola* is now known from 27 species, of which 17 have reliably associated females. Where possible, freshly collected females were dissected to reveal details of the reproductive system, and most were shown to have a female accessory gland, which probably contributes to the materials in the egg covering.

### 3.2. Phylogeny

The results of the phylogenetic tree constructed using the maximum likelihood method and the Bayesian method showed complete consistency, which is consistent with previous studies based on mitogenome and morphological characters [2]. The final analysis results obtained by the two methods were summarised and are presented in Figure 1. From the results, all nodes of the phylogenetic tree obtained high posterior probabilities (Bayesian posterior probabilities > 0.955), although the bootstrap support of the node connecting the *Aquatica* + *Nipponoluciola* branch and (*Luciola* + (*Abscondita* + *Pygoluciola*)) was only 28.8 (Figure 1).

The eleven *Pygoluciola* species in our phylogenetic analysis form a well supported (99.6/1/99) clade, which contains two clearly distinct subclades: Clade A (100/1/100) and Clade B (100/1/100). Clade A contains species from morphological Group 3 (e.g., *P. qingyu*, a large firefly with black elytra), and Clade B contains species from morphological Group 5 (e.g., *P. baise* sp. nov., a small firefly with yellow elytra).

We included the mitogenome sequences of three unidentified *Pygoluciola* species from Genbank (NCBI REF) in our phylogeny (Figure 1): *Pygoluciola* sp. (NCBI Genbank ID: MZ571356) clustered within Clade A (Group 3), the sister taxon of *P. quzhou* sp. nov. within Clade B (Group 5); *Pygoluciola* sp. (NCBI Genbank ID OP747324) from Thailand, the sister taxon of *P. baise* sp. nov.; and *Pygoluciola* sp. FM18 (NCBI Genbank ID: MK292102), the sister taxon of *P. manmaia* sp. nov. However, *Pygoluciola* sp. FM18 and *P. manmaia* sp. nov. were separated by very short branch lengths (0.0003 + 0.0056 = 0.0059), and the collection site of *Pygoluciola* sp. FM18 (Mengla, Yunnan Province, China) is very close to the collection site of the type specimen of *P. manmaia* sp. nov. Therefore, *Pygoluciola* sp. FM18 may be *P. manmaia* sp. nov.

**Figure 1 insects-16-00394-f001:**
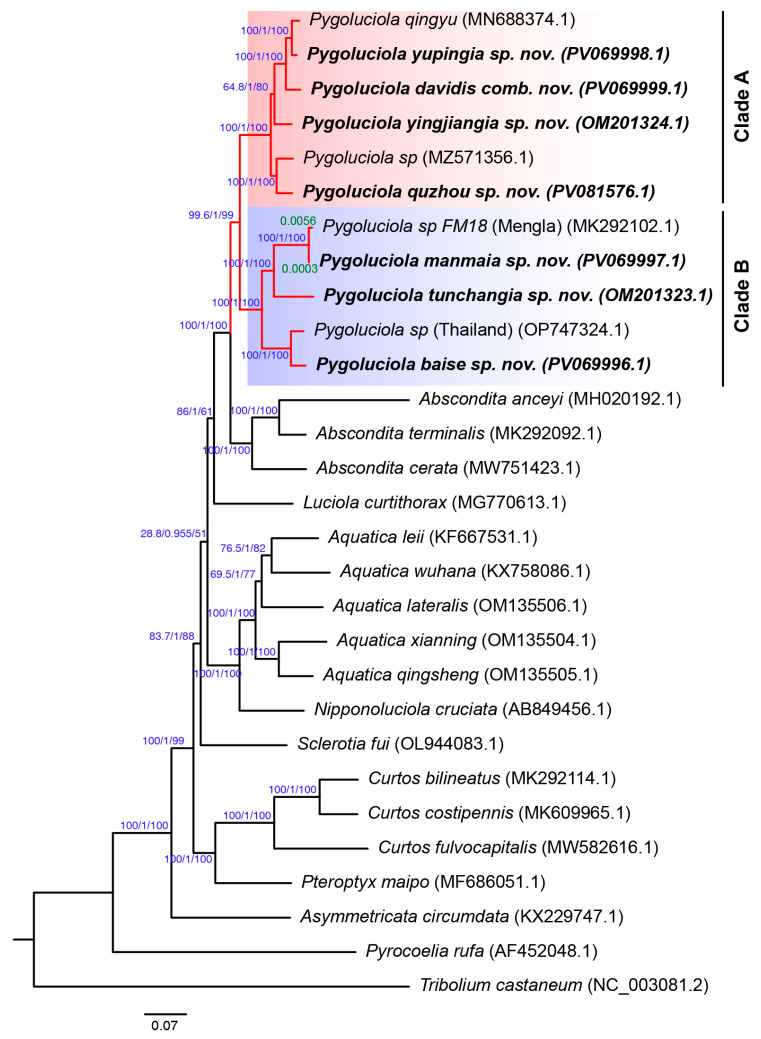
Phylogeny of 28 Luciolinae species inferred from concatenation analysis of 37 mitogenome regions (13 PCGs, 2 rRNA genes, and 22 tRNA genes) using maximum likelihood and Bayesian inference analyses. *Tribolium castaneum* was used as an outgroup. Blue values near internodes correspond to bootstrap support/Bayesian posterior probabilities/SHaLRT test. Green values near the branch are the branch length (genetic distance of two genes). The red branches represent the genus *Pygoluciola*, which contains two sister clades, indicated by the red shade (Clade A) and the blue shade, respectively (Clade B).

### 3.3. Taxonomy

#### 3.3.1. Generic Description


**
*Pygoluciola *
**
**Wittmer, 1939**


See [11] (p. 396) for a full table of synonymies.

**Type species**: *Pygoluciola stylifer* Wittmer, 1939, by monotypy [1] (RMNH).

We present an expanded diagnosis of males, females, and larvae.

**Diagnosis**:

**Male**: Many *Pygoluciola* are difficult to diagnose to genus using only external morphology if they are without the terminal abdomen modifications for which the genus was named, and may require dissection to reveal the diagnostic features of the aedeagus and aedeagal sheath. *Pygoluciola* belongs to that group of Luciolinae genera where males have the aedeagal lateral lobes visible at the sides of the median lobe [2] (key to genera, p. 11; figures 1, 2, 5, 6). It differs from all other such genera, as the lateral lobes are divisible into two distinct sections, a basal sclerotised portion partially separated along its mid-dorsal length, with a strongly asymmetrical anterior margin inclining to the left, and an apical membranous portion probably exserted mainly by fluid pressure [2] (figures 50, 51). Forms of *Pygoluciola* without terminal abdomen modifications resemble *Abscondita* Ballantyne [12], where the aedeagus has a wide sclerotised basal portion with very short posterolateral membranous projections [2] (figures 44–49). The aedeagal sheath is symmetrical, often with an elongate, slender, subparallel-sided sheath sternite expanding gradually towards the posterior end, where it is further expanded and may be apically truncated, medially emarginated, or terminated by a variety of lobes [2] (figure 38). In *Abscondita*, the symmetrical sheath is of a similar form but with sternite expanding evenly along its length [2] (figures 39, 40).

The pronotum is usually wider across its posterior portion (C > A, B), with divergent lateral margins, and subequal to or less than the humeral width (variations in these features addressed below). Interstitial lines are poorly defined, except in *P. cowleyi* and *P. matalangao*. The head is usually small enough to be partly retracted into the prothoracic cavity, with GHW subequal to the width of the prothoracic neck cavity, except in *P. baise* sp. nov., where GHW is slightly larger (Figure 8b). Antennae are always longer than GHW and less than twice GHW (except in *P. dunguna*, where they are longer than 2 × GHW), with all segments elongate filiform; apical labial palpomere flattened, triangular in outline, with inner margin dentate. Abdominal T7 is often concave at the sides, reflecting the torsion of the dorsoventral muscles. Species of *Pygoluciola* exist in five different male forms, which are further explored in detail below.

**Female**: Known females are macropterous. Colour: like male, except for pale LO in V6 only (V7 coloured differently to male—usually pale brown, semitransparent). Pronotum: width less than humeral width. Elytra: contiguous along their sutural margins for almost all their length. Head: close to male head size, with well developed mouthparts and assumed capable of feeding; antennae with elongate slender flagellomeres; apical labial palpomeres laterally flattened, with inner margin dentate. Legs: tibiae of all legs curved in *P. guigliae*, not so in other species. Abdomen (Figure 4e–i, Figure 5d,m,n, Figure 8f–h,r,s, Figure 9f,t,u and Figure 10f,t,u): posterior margin of V7 strongly emarginated; anterior margin of V8 and median apodeme often darker than remainder and may appear as if separated from the posterior area, as intervening membrane is pale coloured; V7 with accessory external developments (transverse ridge) in two species (Figure 2a,b,e and Figure 6f); T7 with anteromedian mound in *P. kinabalua* [20]. An apparent hole between base of V7 and 8 in *P. kinabalua* is an artefact of drying and constriction of strong dorsoventral muscles and is addressed below. Genitalia: with styli longer than wide, parallel-sided, apically rounded, and visible beyond the end of V8; coxites not well differentiated, partly sclerotised, and lateral margins of distal gonocoxites converging posteriorly; valvifers elongate, slender, well sclerotised, and rod like, extending in front of coxites for about ^7^/_10_ of the total length of the complex [35] (figures 35 and 36). Bursa (Figure 2j, Figure 4j–m, Figure 5l, Figure 8q, Figure 9s and Figure 10s): plates a pair of small anteriorly directed hooks, attached only at their bases in *P. guigliae*, *P. satoi*, and *P. wittmeri*. Base of hooks appears split in two, with both arms very elongate slender and reaching into the lumen in *P. dunguna* [7] (figures 34–37); bursa plates very reduced in *P. rammale*, *P. ruhuna* [11] (figures 22, 45) and were not observed in several of the new species described here; several species possess a female accessory gland (e.g., Figure 4j,k).

**Larva**: Larvae are elongate slender with strong well sclerotised dorsal plates, having four blunt spines arising from the margins of all tergal plates but the terminal one, ref. [7] (figure 40) and mandibles with one or two inner teeth; terminal sense organ beside AS3 elongate slender; apical maxillary and labial palpomeres with terminal sense organs. Some are riparian and may be semiaquatic. Very similar to larvae of *Abscondita* and not obviously distinguished—apart from those of *Abs. anceyi*, which have numerous short spines over the dorsal surface—and is the only known species of *Abscondita* larva to have two mandibular teeth [2] (p. 25).

#### 3.3.2. Key to Species of *Pygoluciola* Using Males

Modified and expanded slightly from [2]. For additional figure references, see [11].

Median posterior margin of both V7 and T8 narrowed, prolonged, and often curving, V7 dorsally and T8 ventrally, and at least partially engulfing each other [2] (figures 41, 42), ref. [35] (figures 18–26) with apices of both often emarginated (Group 1)………………………………………………………………………………………………………………………………………………….2Median posterior margin of T8 not narrowed nor curving ventrally; median posterior margin of V7 either narrowed and elongated (Group 2) or median posterior margin of V7 rounded (may have short, slightly developed MPP) (Figure 3b, Figure 4c, Figure 5b, Figure 6d, Figure 7b, Figure 8b, Figure 9d and Figure 10d)………………………………………………………………7All tibiae curved; lateral margins of elytra tapering posteriorly along their length…………………………………………………...3No tibiae curved; lateral margins of elytra usually subparallel-sided, not tapering posteriorly (Figure 3, Figure 4, Figure 5, Figure 6, Figure 7, Figure 8, Figure 9 and Figure 10)……………………………………………………………………………………....4Apex of median posterior projection of T8 no wider than rest and rounded, not emarginate; lateral margins of pronotum slightly sinuate [35] (figures 1, 6, 21)…………………………………………………………………………………....*guigliae* BallantyneApex of median posterior projection of T8 wider than rest and medianly emarginate; lateral margins of pronotum not slightly sinuate [35] (figures 4, 20, 23–25)………………………………………………………………………………………........*stylifer* WittmerMedian posterior projection of V7 bifurcate at apex [35] (figures 18, 26)……...……………………………………….…………........5Median posterior projection of V7 not bifurcate at apex …………………………………………………………………………............6Apex of median posterior projection of V7 deeply emarginate, laterally ensheathing the downturned apex of T8 and projecting laterally beside it…………………………..………………………………………………………………..…………..*wittmeri* (Ballantyne)Apex of median posterior projection of V7 shallowly emarginate, not laterally ensheathing the downturned apex of T8 nor projecting beside it [20] (figures 5, 10) ………………………………………………………………………...........*kinabalua* (Ballantyne)MPP of V7 elongate slender, longer than wide, ventral surface shallowly depressed along its length, bearing on its median dorsal surface two slender teeth; posterior apex of MPP not shallowly depressed; lateral margins of T8 downturned ……………………………………………………………………………………………………………………………….......*satoi* BallantyneMPP of V7 shorter, about as wide as long, ventral surface not shallowly depressed along its length, not bearing dorsal teeth; posterior apex (face) of MPP shallowly depressed; lateral margins T8 not downturned [2] (figures 41, 42); ………………………………………………………………………………………………………………………………..*hamulata* (Olivier)Median posterior margin of V7 prolonged (longer than wide), broad, subparallel-sided, ventral surface shallow, apex slightly emarginated; LO retracted from lateral and posterior margins; T8 with triangular posterior outline [2] (figures 452–455); aedeagal sheath sternite expanding gradually along its length (Group 2)……………………………………………………………...8Median posterior margin of V7 not usually prolonged, if slightly prolonged, often wider than long, and ventral surface not shallow, apex rounded; LO retracted from lateral and posterior margins or not; T8 without triangular outline, usually subparallel-sided, and posterior margin rounded or truncate………………………………………………………………………........9Posterior margin of LO in V7 rounded, without emargination; lateral margins of V7 and MPP pale yellow [7] (figures 2–4,10,11,14)……………………………………………………………………………………………………………………......*dunguna* NadaPosterior margin of LO in V7 with median emargination; lateral margins of V7 and MPP dark brown [2] (figures 449–455)………………………………………………………………………………………………………….……………………..*tamarat* JusohPosterior margin of T8 about as wide as preceding area of T8 and narrowly downturned [2] (figures 439, 440) (Group 3)……………………………………………………………………………………………………………………………………..………….10Posterior margin of T8 usually narrower than preceding area of T8 and not downturned……………………………..………......15V7 with short apically rounded MPP; LO in V7 subparallel-sided, retracted at sides from lateral margins of V7; elytra pale brownish yellow with white apices (underlying fat bodies), pronotum yellowish brown without darker median markings; ventrobasal portion of paired T8 arms with small pointed projections; aedeagus with elongate LL extending well beyond the ML apex; BP narrow; ref. [2] (figures 437–448)……………………………………………………………………………...*phupan* BallantyneMedian posterior margin of V7 rounded, without MPP, or with scarce or ill-defined MPP; LO in V7 with rounded margins occupying almost all of V7; elytra either very dark brown to black, sometimes with paler margins, or semitransparent pale brown with darker brown markings at base; ventrobasal portions of paired T8 arms without projections; aedeagus with short LL often interned at apices, not extending well beyond the ML apex; BP wide, well sclerotised …………………………………………………………………………………………………………………………….………………..……11Elytra black, with or without paler margins; pronotum with paired, well separated, ovoid, brown to black markings; if markings approach closely, then not extending to posterior margin……………………………..…………………………………....12Elytra pale, semitransparent, brown with darker brown markings at base; pronotum with paired median dark brown markings closely approaching in mid-line, with lateral margins divergent and extending to posterior margin (Figure 6) ……………………………………………………………………………………………………………………………….*yingjiangia* sp. nov.Elytra black with pale, semitransparent suture, apex, and lateral margins; MS and MN pale yellow, semitransparent; pronotum with small, paired, ovoid, dark brown well separated markings (Figure 3)………….……………......*davidis* comb. nov.Elytra black without any paler margins; pronotum either marked as above or with extensive median brown markings closely approaching in median line and with divergent lateral margins; MS and MN sometimes with pink fat body undertones……………………………………………………………….……………………………………………………………….........13T8 with paired arms expanded in vertical plane; lateral margins slightly convergent in posterior ¼; LL inturned at their apices; dark markings on pronotum closely approach along midline, with divergent lateral margins and slightly produced at posterolateral corners (Figure 7)…………………………………………………………………………………………...*yupingia* sp. nov.T8 subparallel-sided with paired arms narrow not expanded; LL not inturned at their apices; dark pronotal markings restricted to ovoid well separated areas ……….…………………….…………………….……………………………………………....14LO scarcely retracted from lateral and posterior margins of V7; MPP scarce; metaventrite black; basal abdominal ventrites black (Figure 4)……………………………………………………………………………………………………....*qingyu* Fu & BallantyneLO in V7 retracted from lateral and posterior margin, with short semitransparent MPP; metaventrite pale yellow with scarce brown median markings; basal abdominal ventrites pale brown, posterior margin of V6 narrowly mid-brown (Figure 5)………………………………………………………………………………………………………………………………....*quzhou* sp. nov.Aedeagal sheath sternite terminated by hairy curved (boomerang shaped) projection; aedeagus wider across middle than elsewhere, with membranous apical portion of LL wider at bases and tapering to apices [2] (figures 359–362, 377–379, 401–403, 458–462); (Group 4) …………………………………………………………………………………………………………………………..16Aedeagal sheath sternite not terminated by hairy curved (boomerang shaped) projection, except in *P. nitescens*, where the aedeagus is not wider across middle than elsewhere [2] (figures 422–436); aedeagus subparallel-sided, not wider across middle than elsewhere; membranous apical portions of the LL as wide at their base as at their apex [2] (figures 370–373, 389–393, 413–417, 428, 430–432); (Group 5) ..…………………………………………………………………………………………………....19Elytra dark brown, with brown base, and narrowly orange lateral and sutural margins, which may extend around apex……………………………………………………………………………………………………………………………………………..17Elytra mid-brown, always with base of elytron narrowly to widely paler orange, and narrowly orange lateral and sutural margins, which may extend around apex………………………………………………………..…………………………………….......18Elytron with narrowly pale apex; all abdominal tergites dark brown; LO in V6,7 retracted from lateral and posterior margin of V7, with narrow dark brown margins [2] (figures 374–381)……………………………………………………..*bangladeshi* BallantyneElytron with apex dark brown; T8 paler brown than rest; LO in V6,7 not retracted along lateral or posterior margins, without narrow dark brown margins [2] (figures 456–462)……………………………………….……………………...………….....*vitalisi* (Pic)All abdominal tergites very dark brown; LO in V6,7 not retracted from margins; margins not narrowly dark brown; known only from the Andaman Islands [2] (figures 394–404)…………………………………………………...……………..*insularis* (Olivier)T8 paler brown than rest, semitransparent; LO in V6 narrowly retracted along lateral margins, and along lateral and posterior margins in V7; margins narrowly dark brown; known only from Myanmar [2] (figures 356–363) ….....……...*abscondita* (Olivier)Elytra yellow without darker brown markings, or yellow with apices black......……………………..……..………………………..20Elytra not as above, either dark or pale brown …………………………………………………….………………….……..…………...24Sri Lankan; elytra yellow with no darker markings; aedeagal sheath sternite expanding along its length either in a regular fashion or with lateral margins arcuate…………………………………….…………….…………..……….………….......….....….......21Not known from Sri Lanka; found in mainland China; elytra yellow to orange yellow with some darker markings, especially at apex; aedeagal sheath sternite narrow, subparallel-sided along its length…………………………………………….....……......22Dorsal body yellow, without any underlying pinkish fat body; LO fills V7, except for narrow apical margin; aedeagal sheath sternite apically truncate with small, narrow median emargination; LL of aedeagus not expanded in horizontal plane [11] (figures 2, 3, 6–19)…………………………………………….………………….…………….……………..*rammale* Wijekoon & De SilvaDorsal body yellow with areas of underlying pinkish fat body; LO restricted to median area in V7 and retracted from lateral and posterior margins, sides of LO tapering slightly; aedeagal sheath sternite apically rounded without median emargination; LL of aedeagus wide, expanded in horizontal plane [11] (figures 25, 26, 29–42)………………………………………….………………….………………….…………..……………….....*ruhuna* Wijekoon & De SilvaLO in V7 with posterior margin emarginated (Figure 8)………………………………….……….………………….….....*baise* sp. nov.Posterior margin of V7 LO not emarginated………………………………………….………………………….……....….……...….…23Basal abdominal ventrites black (Figure 10)…………………………………….….………….………………….…...*tunchangia* sp. nov.Basal abdominal ventrites yellow (Figure 9)………………….…………………….…….………..………………….....*manmaia* sp. nov.Elytra dark brown to black without paler margins………………………………………….……………….………………….…….....25Elytra never dark brown without pale margins; either mid-brown with all margins orange yellow, or mid-brown with lateral margin paler brown, or elytra brown with only suture paler brown………………..………………………………..………………..2612.0 mm long; heavy bodied; LO in V7 retracted from posterior margin; MPP of V7 short, apex broadly rounded; T8 parallel-sided; LL broadly expanded in horizontal plane, not much longer than ML [2] (figures 422–436)……..…….....*nitescens* (Olivier)9–10 mm long; slender bodied; LO in V7 parallel-sided, retracted from lateral and posterior margins; MPP of V7 broad, apically truncate; T8 not parallel-sided; LL not broadly expanded in horizontal plane, elongate slender, about 2 X length of ML [2] (figures 382–393)…………………………………....……………..………….……...…..………...……………..…*calceata* (Olivier)Indonesian (Java); elytra mid-brown with lateral, sutural, and apical margins orange; V7 without reflexed margins [2] (figures 364–373)……………………………………………………………………...………....……..………………..……...……….*ambita* (Olivier)Either from Australia or the Philippines; with lateral margin only paler than rest, or elytra brown with suture paler than rest……………………………….………………………………………………..………..………………...………………………....…..….27Australian (Northern Territory around Darwin); large exposed head with posterior eye emarginations; pronotum with subparallel sides, wide median dark marking reaching to both anterior and posterior margins; elytra mid-brown with paler lateral margins; margins of V7 are not reflexed……………………..………..……………..……………………......*cowleyi* (Blackburn)Philippines; head not large, exposed, nor with emarginations; pronotal margins not subparallel-sided; elytra yellowish–mid-brown; reflexed margins of V7 envelop T8 at sides [2] (figures 405–421)………..…………………………......*matalangao* Ballantyne

#### 3.3.3. Key to Females of *Pygoluciola*

All tibiae curved………………………………………………………………………………....………………………*guigliae* (Ballantyne)No tibiae curved………………………………………………………………………………....…………………………………………….2Dorsal colour yellow, without darker brown markings; bursa hooks minute……………………………..…………………………. 3Dorsal colour always with some darker brown to black markings, on pronotum, elytra; bursa hooks, if observed, large, well defined……………………………………………………………………………………………………………………………………..…....4V2 light brown; V3 dark brown; V4,5 black; no underlying pink fat body evident [11] (figures 4, 5)………………………………………………………………………………………..………………………*rammale* Wijekoon & De SilvaV2−4 dark brown, 5 black; pink fat body visible beneath cuticle of pronotum andelytral apices [11] (figures 27, 28)………………………….…………………………..……….…..………..*ruhuna* Wijekoon & De SilvaElytra black, with or without paler margins …………………………………………………………………………….…………........…5Elytra never uniformly black; pale brown with apices black, or pale brown with darker brown at base...….....…….…....…….…6Black metaventrite, and abdominal V2-5; posterior emargination of V7 reaching beneath the posterior margin of V6 (Figure 4d–m)…………………………………………... ………………………………………….………………………....*qingyu* Fu & BallantynePale brown metaventrite and abdominal V2-5; posterior emargination of V7 not reaching beneath the posterior margin of V6 (Figure 5c,d,k–n)…….…………………………………………………………………………………..……………………..*quzhou* sp. nov.Dorsal surface pale, semitransparent, light brown, or yellow, with dark brown to black elytral apices………………………………………….……….…………………………………………………………………...…………………....7Dorsal colouration never as above……………………………………..……………………………………....……………………….…..10Dorsal surface very pale brown, semitransparent, with brown elytral apices; ventral body pale yellowish brown, with narrow dark brown band across posterior margin of V5; each bursa hook in two elongate curved sections [7] (figures 25–37)………………………………………………………………………………...………………….…………………………...*dunguna* NadaDorsal surface yellow with black elytral apices; ventral body either pale yellow with minimal darker markings or V2-5 very dark brown; bursa plates not detected (but absence not assumed)……………………………..………………………………………..8V2-5 and metaventrite yellow with no dark brown markings; anterior corners of T7 broadly rounded (Figure 9e,f,r–u)……………………………….………………………………………..…………….....…………………….......…………*manmaia* sp. nov.V2-5 and metaventrite dark brown to black; anterior corners of narrowed, acute………………………….……….……………....…9Posterior emargination of V7 not projecting beneath LO in V6; posterior corners of V7 rounded; anterior corners of T7 narrowed acute; anterior apodeme of V8 separate from posterior area (Figure 10e,f,r–u)……………..………..*tunchangia* sp. nov.Posterior margin of V7 deep, projecting beneath LO in V7; posterior corners of V7 acute; anterior corners T7 elongate slender, acute, slightly divergent; anterior apodeme of V8 continuous with posterior area (Figure 8e–h,p–s)………………...*baise* sp. nov.Posterior margin of V7 deeply emarginate; bearing a small ridge anterior to median area of deepest emargination (possibility this ridge is an artefact, having been seen only on dried pinned specimens); V8 with anteromedian prolongation not any more sclerotised than remainder of V8; T7 with anteromedian area rounded and elevated, lateral areas not flattened [20] (figures 16, 17); (Figure 2)…………………………………………………………………………....……………………………...*kinabalua* (Ballantyne)Posterior margin of V7 with or without an anteromedian ridge; V8 with anteromedian prolongation well sclerotised and visibly separated from remainder of V7; T7 without a rounded and elevated anteromedian area, with lateral areas flattened……………………………………………….………………………………………………………………………………………..11V2-5 very dark brown to black; V7 with anteromedian ridge narrowly edged in black; elytra pale, semitransparent, light brown with dark brown markings across base (Figure 6e,f)…………………………….…….…………………….*yingjiangia* sp. nov.V2-5 yellowish to mid-brown, never black; V7 without anteromedian ridge; elytra mid-brown, with suture or suture and lateral margin semitransparent and paler than rest; without darker brown markings at elytral base……………………..…………..………..………………..………………..……………………………..………………..………………12Pronotum with extensive median dark brown markings reaching to posterior margin; elytra mid-brown with suture narrowly yellow; posterolateral areas of V7 irregularly expanded, with aggregation of fat body beneath ……………………………………………………………………….………………..………………..…………..…....……*satoi* (Ballantyne)Pronotum with paired ovoid brown markings not attaining posterior margin; elytra mid-brown semitransparent, with both suture and lateral margin paler brown than rest; posterolateral areas of V7 not expanded, extent of fat body distribution not determined in original examination………………………………..………………..………………..…………...…*wittmeri* (Ballantyne)

#### 3.3.4. Overview of Species of *Pygoluciola* Including Descriptions of New Species


**List of species of *Pygoluciola:* see Appendix A.**


*Pygoluciola* males exist in five different forms, as follows (expanded and modified from [11]).

**Group 1**: Males: Six species resemble the form originally described by [1]. Body elongate slender, L/W 3–3.4. Pronotum: approximately 1/10 body length; with divergent straight lateral margins (C < A, B) and angulate corners; width subequal to humeral width, less in P. satoi. Elytron: interstitial lines not well elevated. Abdomen: LO in V7 retracted from lateral and posterior margins; V7 observed in some species, with margins outside the LO narrowly uprolled onto the dorsal surface; with slender median posterior prolongation of V7 curving dorsally at its apex, often engulfed by a similar slender median prolongation of T8; T8 subparallel-sided along most of its length. Aedeagus: elongate, LL long; ML short, except in *P. wittmeri* and *P. kinabalua* (LL short, ML long). Aedeagal sheath: sternite elongate slender, subparallel-sided, extending well posterior to lateral tergite articulations before expanding at its apex.

Females: macropterous with hooked bursa plates; lateral and posterior margins of V7 uprolled onto dorsal surface in some species; posterior margin of V8 deeply emarginated; V8 with elongate slender apodeme usually separated by membrane from posterior area; T7, T8 with narrowed apically acute anterior corners in *P. kinabalua*. Female reproductive system not investigated in detail; *P. kinabalua* known to have accessory glands.

Larvae: not associated.


**
*Pygoluciola guigliae *
**
**(Ballantyne)**


For taxonomic and figure references to this species, see [2] (p. 121) and Appendix A.

**Type**: Male. British North Borneo Bundu Tukan (NHML).

**Diagnosis**: Male. 11–11.9 mm long; one of two species with curved tibiae [36] (figure 118), distinguished from *P. stylifer* by sinuate lateral pronotal margins (*P*. *stylifer* margins not sinuate) [35] (figures 1, 4), entire rounded apex of T8 (*P. stylifer* expanded, shallowly emarginated); MPP apex not emarginated; aedeagus [35] (figures 1, 6, 13–17); ref. [36] (figures 109, 111–112, 116–118, 120, 121, 124, 125); without accessory plate in association with tergite [35] (figures 15–17).

Female. 12 mm long; macropterous observed in flight; all tibiae curved; posterior and lateral margins of V7 uprolled onto dorsal surface; T8 with anterior apodeme separated; ovipositor elongate slender; bursa plates short hooks [35] (figures 30, 32, 34, 35). Reproductive system described [35] (figure 37 is a diagrammatic representation of the reproductive system and has P and S transposed (S = spermatophore digesting gland; P = spermatheca). No investigations made into nature of T7 or anterior corners of V7.

Larva not associated.


***Pygoluciola hamulata* (Olivier)**


For taxonomic and figure references to this species, see [2] (p. 121), ref. [35] (p. 24), ref. [37] (p. 3), and Appendix A.

**Lectotype**: Male. Borneo, Sarawak (MCSN).

**Diagnosis**: Male. 8.5 mm long (lectotype); tibiae of legs not curved; T8 apex rounded, MPP with expanded posterior face, neither apex emarginate; aedeagus [2] (p. 121, figures 41, 42); ref. [36] (figures 113, 114, 126–129); ref. [37] (p. 3, figures 1, 2). Aedeagal sheath not investigated.

Female and larva not associated.


***Pygoluciola kinabalua* (Ballantyne)**



Figure 2


For taxonomic and figure references to this species, see [20] (p. 371).

**Type**: Male. **Malaysia**, Sabah (NHML).

**Specimens examined**: Malaysia: Sabah. 6.10 N, 116.40 E, Mt. Kinabalu, Mesialu (camp): 5000 ft, 13–15.iii.1964, S. Kueh, three females (taken with three males); five females, J. Smart [31.i.1964, female; 2.ii.1964, female; 6.ii.1964, female (taken with one male); 19.ii.1964, two females (taken with one male)] (NHML).

**Diagnosis**: **Male.** 10.6–11.3 mm long; tibiae not curved; MPP apex emarginated; apex of T8 slightly biemarginated; aedeagus LL short, ML long [38] (figures 13–15); without accessory plate in association with tergite of aedeagal sheath.

**Female**. 10.9–12.0 mm long; distinguished from other Luciolinae females by the ridge in the median area anterior of the V7 emargination, seen only in one other species *P. yingjiangia* sp. nov., and the apparently unique swollen area on the surface of T7 (Figure 2c). Female abdominal modifications were described in [20]; pinned female specimens were dissected in an attempt to further clarify their nature.

Colour: pronotum light brown with some median dark markings or yellow with no darker markings; elytra brown. Abdomen (Figure 2a–k): no elements of reproductive system detected beneath the mound on T7 (Figure 2c,f–i); the hole perceived by [20] at the base of V7 is considered an artefact of dehydration and the tension of the dorsoventral muscles (Figure 2d); anterior corners of both T7,8 narrowly elongated; anterior corners of V7 irregularly and narrowly produced. Reproductive system: minute bursa plates detected (Figure 2j,k); possible FAG present (Figure 2g).

**Discussion**: The possibility of a copulation clamp in *Pygoluciola kinabalua* was not completely discounted in [20], where it was the anatomical features of the female abdomen that suggested a possible clamp, with the suggestion that the apex of the male T8 engaged against a hard swollen surface on the female T7 (Figure 2c,f), while the tip of male V7 would abut against the median ridge in the anteromedian portion of the female V7 (Figure 2a,b) [20], once the pair were coupled and turned tail to tail. Subsequently [35] (p. 45), the suggestion was explored for other species of *Pygoluciola*, where ethanol preserved specimens showed a wide divergence between the terminal abdominal segments. The strong dorsoventral musculature was evidenced by lateral depressions of T7. No further modifications in the female observed here support these suggestions. The opening at the base of the female V7 was considered an artefact of dehydration (Figure 2d); ref. [37] (p. 7).

Reexamination and dissection of pinned paratype females revealed only an egg accumulation under the mound on T7 and minute paired hooked bursa plates. Our findings confirm that the hole is a post-mortem artefact with no function. However, the depressed areas on V7 and T7 suggest strong dorsoventral muscle attachment, which is probably reflected in the extra surface area seen at the anterior corners of T7,8. The environmental context for these females remains unknown.

**Figure 2 insects-16-00394-f002:**
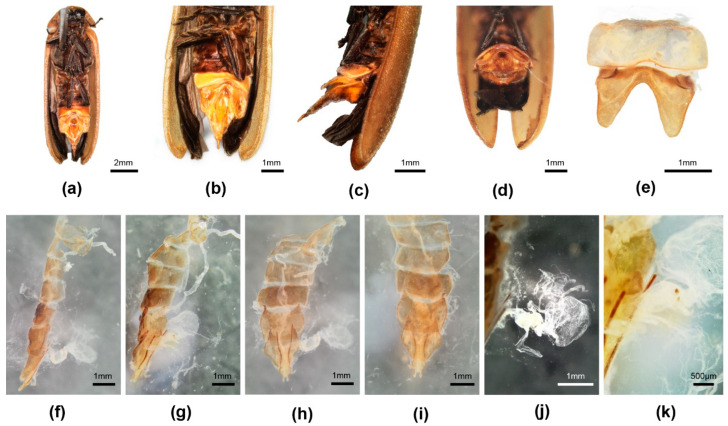
*Pygoluciola kinabalua* paratype female. (**a**) ventral habitus; (**b**) ventral abdomen; (**c**) left lateral abdomen; (**d**) end of abdomen from behind, ventral surface uppermost; (**e**) dorsal V6,7; (**f**–**i**) dissected abdomen, tergites only: (**f**) right lateral; (**g**) slightly oblique view of f; (**h**) ventral; (**i**) dorsal; (**j**) reproductive system, spermatophore digesting gland to right; (**k**) detail reproductive system showing minute bursa plates.


**
*Pygoluciola satoi *
**
**Ballantyne**


For taxonomic and figure references to this species, see [37] (p. 4).

**Type**: Male. **Philippines** (ZRC).

**Diagnosis**: **Male.** 8.8–10.2 mm long; tibiae not curved; apices of MPP and T8 rounded, not emarginated; MPP with elevations along dorsal surface [37] (figures 5, 6); aedeagus LL long, ML short; aedeagal sheath with accessory plate in association with tergite [37] (figures 8, 9).

**Female.** 9.5 mm long; macropterous observed in flight; coloured as for male; no tibiae curved; dorsal surface V7 and anterior corners of T8 not determined in original investigation; ovipositor similar to that of *P. guigliae*; bursa plates short hooks [37] (figure 7).


**
*Pygoluciola stylifer *
**
**Wittmer**


For taxonomic and figure references to this species, see [1,2,35].

**Type**: Male. **Indonesia**, Long Petah (RMNH).

**Diagnosis**: **Male.** 11.2 mm long. Most similar to *P. hamulata*, distinguished by non-curved tibiae and T8 apex emarginate (*P. hamulata* has all tibiae curved, T8 apex rounded entire) [35] (figures 4, 20, 23–25); ref. [36] (figures 107–108, 110, 115, 119, 122–123), aedeagus [36] (figures 122–123), ref. [38] (figure 6).

Female and Larva not associated.

**Remarks**: Without a direct observation of the holotype [35,36], legs were recorded as straight (comparisons were made by the curator at CMG). Subsequent examination of the type permitted reassessment of the interpretation of the shape of the tibiae.


**
*Pygoluciola wittmeri *
**
**(Ballantyne)**


For taxonomic and figure references to this species, see [3,4,5,35,36,37].

**Type**: Male. British North Borneo (BPBM).

**Diagnosis**: **Male.** 7.8–10.1 mm long. Distinguished from all other species in Group 1 by the widely emarginated V7 apex ensheathing tip of T8 at sides [3] (figures 4, 5), ref. [4] (figure 3g,h), ref. [5] (figure 2), ref. [35] (p. 33, figures 5, 18, 22, 26); dorsal colouration light brown, pronotum with paired medially separated brown marking, elytral humeral angles pale brown and apices reddish brown; all tibiae straight; aedeagus short, LL short with apices inturned, ML long [36] (figures 136–142), ref. [35] (figures 8, 9); aedeagal sheath without accessory plate in association with tergite.

**Female.** 11.5–12.0 mm long. Macropterous observed in flight. Coloured as for male; no tibiae curved; posterior and lateral margins of V7 uprolled onto dorsal surface [35] (figure 31); T8 anterior apodeme not clearly separated; ovipositor elongate slender; bursa plates short hooks [35] (figures 28, 29, 31, 33). Ovipositor elongate slender [35] (figure 36).

**Group 2:** Male: Two species with modified terminal abdomen. Body short, wide (L/W 2.5–2.9). Pronotum: 0.17, as long as BL; lateral margins divergent (C > A, B), all corners broadly rounded; pronotal width subequal to humeral width. Elytron: interstitial lines not well defined. No tibiae curved. Abdomen: V7 having a broad apically truncate median prolongation slightly inclining dorsally along its length in pinned specimens, engaging at its tip against the underside of T8 [2] (figures 450, 451), ref. [7] (figures 10, 12, 13); V7 apex not upturned or expanded; lateral uprolled margins of V7 strongly developed in both species [2] (figure 451); ref. [7] (figure 11); T8 without prolonged slender median posterior area (posterior margin of T8 triangular), lateral wing-like projections at sides, and paired anterolateral arms diverging anteriorly (arms with bifurcate basal area in *P. tamarat*) [7] (figures 12–16); ref. [2] (figures 450, 451, 455). Aedeagus: elongate, LL long, ML short [2] (figures 452, 453); ref. [7] (figures 19–21). Aedeagal sheath: sternite, either elongate slender subparallel-sided along its length before terminal expansion (*P. dunguna*) [7] (figures 22–24) or expanding evenly along its length before terminal expansion (*P. tamarat*) [2] (figure 454); posterior margin truncate with paired hairy lobes.

Female: Macropterous females of *P. dunguna* have paired hooked bursa plates with elongated arms [7] (figures 34–37).

Larva: *P. dunguna* larvae are semiaquatic with bidentate mandibles [7].


**
*Pygoluciola dunguna *
**
**Nada**


For taxonomic and figure references to this species, see [2,7] and above.

**Type**: Male. **Malaysia** (MZUM).

**Diagnosis**: **Male.** 9.1–10.7 mm long. Dorsally dingy brown (in pinned specimens) or bright clear yellow (in ethanol preserved specimens), with diffuse darker brown pronotal markings [7] (figures 2–9); LO with slight median posterior emargination, semitransparent pale yellowish lateral margins of V7, MPP [7] (figures 3, 6, 10, 14); T8 pale brown semitransparent, base of anterolateral arms not bifurcated [7] (figure 15), ref. [2] (figure 455); aedeagal sheath sternite elongate subparallel-sided before apical expansion; (distinguished from *P. tamarat* by the LO with median posterior emargination well defined; lateral margins of V7, MPP dark brown; T8 very dark brown, base of paired arms bifurcated; aedeagal sheath sternite elongate expanding along its length before apical expansion) [7] (figures 22–24).

**Female**. 9.2–10.3 mm long. Macropterous observed in flight. V7 dorsal surface not investigated. Anterior margin of V8 narrowly prolonged [7] (figures 27, 30, 33). T8 not investigated.


**
*Pygoluciola tamarat *
**
**Jusoh**


For taxonomic and figure references to this species, see [2,7] and above.

**Type**: Male. **Thailand** (MZUM).

**Diagnosis**: **Male.** 10.0 mm long. Dorsally dingy brown with diffuse darker pronotal markings; distinguished from *P. dunguna* above [2] (figures 449–455).

**Group 3**: Males: Six species, including two new and one transferred from *Luciola* with typical genitalic features as described for Group 1, but without the posterior margin of T8 narrowly downturned, narrowly prolonged; posterior margin of V7 bluntly rounded, without an elongated median projection, or a short apically rounded MPP, Body elongate L/W 3/1. Pronotum: approximately 1/6 body length; with lateral margins divergent (C > A, B) and all corners rounded (less than 90°); except in *P. phupan*, where B > A, C; width subequal to or less than humeral width. Elytron: interstitial lines not elevated. Abdomen: V7 with median posterior margin either evenly rounded or produced into a short apically rounded horizontal MPP; lateral uprolled margins of V7 not specifically investigated; T8 either subparallel-sided along its length, or lateral margins slightly arcuate, or lateral margins in posterior area gently converging; posterior margin narrowly downturned across its width; paired arms of T8 not bifurcate at base, sometimes diverging anteriorly, much longer than posterior entire portion, except in *P yingjiangia* sp. nov., where they are shorter. Aedeagus: short, squat (L/W 2.2) with LL short, often inturned at their apices, and ML long, or aedeagus elongate, LL long, ML short; BP narrow in *P. phupan*. Aedeagal sheath: sternite elongate slender subparallel-sided before apical expansion.

Female macropterous; bursa where investigated with paired hooked plates.

Larva. *P. qingyu* larvae are semiaquatic; some with bidentate mandibles.


**
*Pygoluciola davidis *
**
**(Olivier) comb. nov.**



Figure 3


For taxonomic and figure references to this species, see [10,39,40].

**Type**: Male. **China.** Kiang-Si Abbi David (MNHN).

**Specimens examined**. **China.** Collector is Fu, X. H. Guangdong Province, Qingyuan City, 16.vi. 2022, three males (HZMAU).

**Diagnosis**: Similar to *P. qingyu*; distinguished by the pale brown median pronotal markings, and the wide semitransparent elytral lateral margin (*P. qingyu* has dark brown pronotal markings and black non-margined elytra).

**Male.** Length 11–12 mm. Colour: (Figure 3a,b) pronotum pale creamy white, semitransparent, with underlying fat bodies visible retracted across median anterior area (black head visible beneath) and irregularly across median posterior area; centre of disc with paired ovoid mid-brown markings; MN whitish due to underlying fat bodies; MS semitransparent, pale creamy white; elytra very dark brown almost black, with suture narrowly pale, semitransparent (colour like MS), pale whitish part of humeral angle, and all of lateral margin and apex widely pale white semitransparent; head and palpi black; antennae with scape pedicel and FS1 black, FS 2-4 with median dark brown areas, remainder creamy white; venter of pro and mesothorax pale faint yellow, of metathorax black, except for a thin posterior yellow brown margin; metepipleural plates brown yellow; legs 1 coxae trochanter and base of femora pale whitish, apical half femora irregularly marked in brown, tibiae and tarsi black; legs 2 coxae trochanter and femora whitish, tibiae and tarsi black; legs 3 coxae pale brown with median brown mark at inner base, trochanters, femora, basal 2/3 tibiae whitish, apical 1/3 tibiae and all of tarsi dark brown; abdominal ventrites black, except for white LO in V6,7 reaching all but narrow lateral and posterior margins of V7; T8 pale brown semitransparent, T7 paler brown than 8.

Pronotum. Lateral margins straight (C > A, B); width subequal to humeral width. Elytron: subparallel-sided with faintly defined interstitial lines. Head: GHW/SIW 3/1; GHW/pronotal width 0.56; head width slightly less than width across prothoracic cavity, allowing head to be partially retracted within; ASD >ASW; antennae longer than GHW but shorter than 2 × GHW. Abdomen: LO occupies all of V7, except for a narrow semitransparent lateral and posterior margin; posterior margin V7 broadly rounded, no MPP present; T7 (Figure 2h) with lateral depressed areas; T8 (Figure 2i) with posterior margin slightly and broadly emarginate; posterior corners slightly produced laterally and rounded; narrowest across posterior 1/4, then margins diverge anteriorly. Aedeagal sheath (Figure 2f,g): apical expansion terminated by paired hairy lobes. Aedeagus (Figure 2c–e): short broad, subparallel-sided, ML short elongate ovoid, LL short inturned slightly at their apices; BP well defined, broad, separated in midanterior margin.

Females and larvae not associated.

**Figure 3 insects-16-00394-f003:**
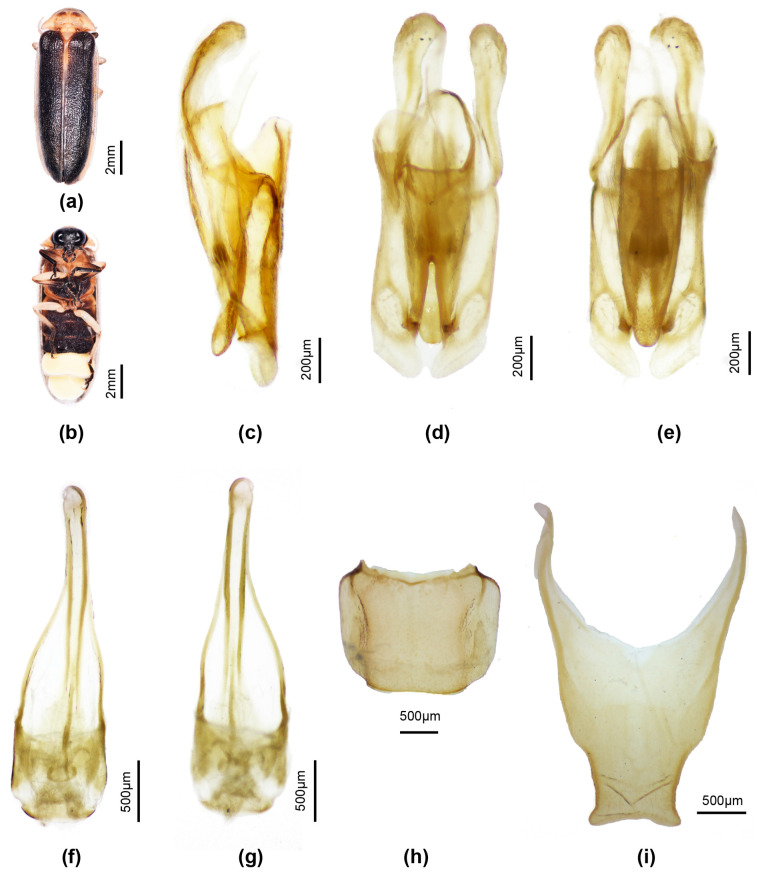
*Pygoluciola davidis.* Male (**a**,**b**). (**a**) Dorsal habitus; (**b**) Ventral habitus. (**c**–**g**) Aedeagus: (**c**) left lateral; (**d**) dorsal; (**e**) ventral. (**f**,**g**) Aedeagal sheath: (**f**) dorsal; (**g**) ventral. (**h**) Tergite 7. (**i**) Tergite 8.


**
*Pygoluciola phupan *
**
**Ballantyne**


For taxonomic and figure references to this species, see [2].

**Type**: male. **Laos** (NHML).

**Diagnosis: Male**. 12.0 mm long; light brown dorsally, without darker brown markings on pronotum or elytra, with pinkish fat body visible beneath pronotum; distinguished from P. yingjiangia sp. nov. by the brown markings at elytral base and extensive median pronotal markings, and from the remaining species in Group 3 by the pale elytral colour (other species have black, sometimes paler margined, elytra) elytra convex sided (B > A, C), aedeagus with BP narrow, ML short and elongated LL. Females and Larvae not associated.


**
*Pygoluciola qingyu *
**
**Fu & Ballantyne**



Figure 4


For taxonomic and figure references to this species, see [8].

**Type**: Male. **China** (NHMHAU).

**Specimens examined: China**. Collector is Fu, X. H. Anhui Province, LiuAn City, 25.vii. 2013, two males; Guangdong Province, Meizhou City, 29.iv.2019, four males; Guangdong Province, Shenzhen City, 24.vi.2016, two males; Hubei Province, Huanggang City, Mt. Dabie, 15.vii.2015, three males; Hubei Province, Xianning City, Mt. Jiugong, 8.vii.2024, six males; Hubei Province, Shiyan City, 3.viii.2019, four males; Jiangxi Province, Ganzhou City, 26.vii.2019, four males; Sichuan Province, Mt. Emei, 15.vii.2018, five males; Sichuan Province, Mt. Yuping, 20.vii.2024, three males; Zhejiang Province, Quzhou City, Kaihua County, 17.vii.2024, four males (HZMAU).

**Diagnosis: Male**. One of three species in Group 3 having black elytra, most obviously distinguished from P. davidis comb. nov. by the wide semitransparent lateral elytral margin in davidis; differs from *P. quzhou* sp. nov. by the ventral colouration (*P. qingyu* has metaventrite and abdominal ventrites in front of the LO black, *P. quzhou* has metaventrite yellow with scarce median brown markings, and abdominal ventrites pale brown). Females with bursa hooks, and larvae associated [8].

**Female**. Length 12–14 mm. Distinguished from *P. quzhou* sp. nov. female by the dark brown ventral colouration (*P. quzhou* sp. nov. has pale brown V2-5 before the white V6 LO. Colour (Figure 4d,e,h): as for male, except for white LO in V6 only, and orange yellow V7,8; V7 with white fat body visible; V8 without underlying fat body. Abdomen (Figure 4e,g–i): posterior margin of V7 with deep emargination not reaching under posterior margin of V6; posterior corners V7 rounded. T8 (Figure 4g) elongate longer than wide, with anterior corners acute, broad at their base; V8 (Figure 4i) with separate anterior apodeme, median posterior margin shallowly emarginated. Reproductive system (Figure 4j–m): bursa with well developed paired hooks; stalked FAG.

**Flash patterns**. Luminous behaviour of both male and female described in [8] (p. 12). Males can synchronise their 0.12 s flash.

**Figure 4 insects-16-00394-f004:**
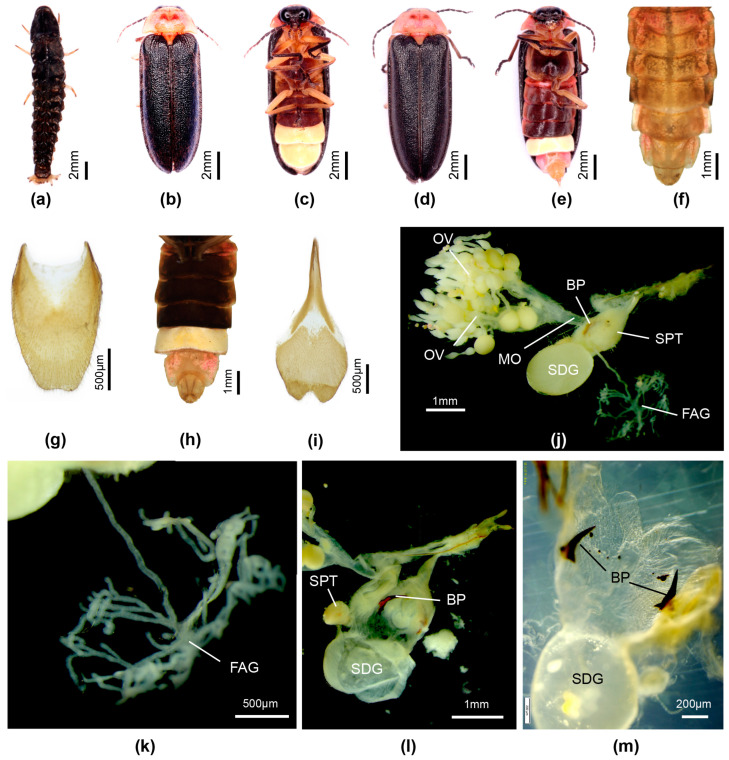
*Pygoluciola qingyu*. Larva (**a**), male (**b**,**c**,**f**), female (**d**,**e**,**g**–**m**). (**a**,**b**,**d**,**f**,**g**) Dorsal habitus; (**c**,**e**,**h**,**i**) Ventral habitus. (**j**–**m**) Female reproductive system: (**g**) Tergite 8; (**i**) Ventrite 8; (**j**) entire female reproductive system before clearing; (**k**) female accessory gland; (**l**) ovipositor (top right) with bursa, BP, SDG, and SPT; (**m**) paired hook-like bursa plates. Legend: BP, bursa plates; FAG, female accessory gland; MO, median oviduct; OV, ovaries; SDG, spermatophore digesting gland; SPT, spermatheca.

***Pygoluciola quzhou* sp. nov**.


Figure 5


**Types**: Holotype. Male. **China**. Collector is Fu, X. H. Zhejiang Province, Quzhou City, Kaihua County, 18.vii. 2024. Paratypes: three males, one female, same data as for holotype (HZMMAU).

**Etymology**: The species is named for its type locality Quzhou City, and the anglicised name is retained without modification and considered here as a noun in apposition.

**Diagnosis**: One of two species in Group 3 having black non-margined elytra; most similar to *P. qingyu*, distinguished by the ventral colouration (*qingyu* has metaventrite and abdominal ventrites in front of the LO black, *quzhou* has metaventrite yellow with scarce median brown markings, and abdominal ventrites pale brown). Females with bursa hooks. Larvae unknown.

**Male**. Length 10–12 mm. Colour (Figure 5a,b): pronotum, MS, and MN semitransparent whitish with underlying pink fat body, and paired small well separated ovoid brown markings ion pronotal disc; elytra black; head antennae and palpi black; undersurface prothorax pinkish; ventral thorax very pale brown with pink markings between coxae 1; metaventrite pale yellow with diffuse median pale brown markings; legs with light brown coxae and trochanters, femora 1 basal half pale brown, apical half diffuse black; femora 2, 3 as for 1 but with basal ¾ pale brown; all tibiae and tarsi black; basal abdominal ventrites pale brown, posterior margin V5 narrowly darker brown; abdominal tergites semitransparent with pink fat body visible beneath;

Pronotum. Lateral margins straight (C > A, B); width slightly > humeral width. Elytron: interstitial lines evanescent. Head: GHW/SIW 4.8; ASD slightly < ASW. Abdomen (Figure 5b): LO retracted from lateral and posterior margins of V7, with broad short irregularly margined MPP discernible; T7 (Figure 5i) with depressed lateral margins; T8 (Figure 5j) posterior margin evenly truncated or slightly emarginated; lateral margins converging slightly in posterior 1/5; anterior arms diverging slightly, expanded in vertical plane. Aedeagal sheath (Figure 5g,h) sternite terminated by paired hairy lobes. Aedeagus (Figure 5e,f): elongate slender; BP broad, well sclerotised, halves adjacent in anterior midline; ML short, apex acute, LL short.

**Female**. Length 11–12 mm. Colour (Figure 5c,d): as for male, pink fat body irregularly scattered beneath prothorax; pink fat body visible in hypomeron, between coxae 1, 2, abdominal V2-5, and sides of V7; basal abdominal V2-5 pale brown. Abdomen (Figure 5d): posterior margin of V7 with deep emargination not extending under posterior margin of V6; V7 with posterior corners rounded; V8 (Figure 5n) with anterior apodeme separated from remainder, median posterior margin shallowly emarginated; anterior corners of T8 (Figure 4k,l) short, narrowly extended, apically acute; posterior margin not emarginated. Reproductive system (Figure 5k,l): well defined paired bursa hooks present; FAG well defined.

Larva unknown.

**Figure 5 insects-16-00394-f005:**
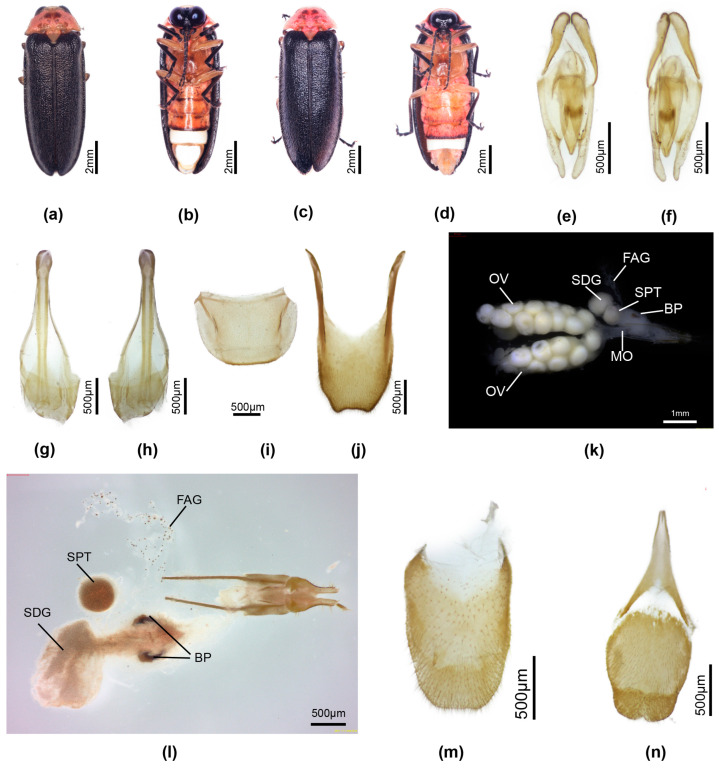
*Pygoluciola quzhou* sp. nov. Male (**a**,**b**,**e**–**j**), female (**c**,**d**,**k**–**n**). (**a**,**c**) Dorsal habitus; (**b**,**d**) Ventral habitus. (**e**–**g**) Aedeagus: (**e**) ventral; (**f**) dorsal; (**g**,**h**) Aedeagal sheath: (**g**) dorsal; (**h**) ventral; (**i**) Tergite 7; (**j**) Tergite 8. (**k**–**m**) Female reproductive system: (**k**) entire female reproductive system before clearing; (**l**) ovipositor with bursa, BP, SDG, SPT, MO, and FAG; (**m**) tergite 8; (**n**) ventrite 8. Legend: BP, bursa plates; FAG, female accessory gland; MO, median oviduct; OV, ovaries; SDG, spermatophore digesting gland; SPT, spermatheca.


**
*Pygoluciola yingjiangia *
**
**sp. nov.**



Figure 6


**Types**: Holotype. Male. **China**. Collector is Fu, X. H. Yunnan Province, Yingjiang County, 23.vii. 2022. Paratypes: two males, one female, same data as for holotype (HZMAU).

**Etymology**: The anglicised name for the locality, Yingjiang County in Yunnan Province, is latinised and considered a noun in apposition.

**Diagnosis**: Distinguished from the only other paler brown species in Group 3, P. phupan, by the darker brown markings on the pronotum and brown markings at the elytral base (P. phupan has no darker markings). Females coloured as for male, with deeply incised posterior margin of V7. Larvae are elongate slender with dorsal plates well sclerotised; thoracic terga have two short acute projections along posterior margins of terga to either side of the median line, and elongated slender apically pointed projections at the posterolateral corners and the posterior area of the lateral margins; abdominal terga 1–8 have a set of four elongated apically acute projections arising from either side of the median line and at the posterolateral corners.

**Male**. Length 13–15 mm. Colour (Figure 6c,d): pronotum pale semitransparent yellow; fat body retracted along anterior margins reveals black head beneath; with dark brown paired median markings separated by a pale median line (except in anterior 1/3), extending to posterior but not anterior margin, wider across posterior margin, with lateral margins slightly irregular and divergent; MS, MN pale yellow, MS with minute anterior median dark brown marking; elytra yellow, semitransparent, underlying darker body and hind wings can confuse interpretation of colour; with dark brown marking at base, marking extending along suture for 1/3 elytral length, not extending to humerus or along lateral margin; head between eyes, antennae and palpi black; ventral surface of thorax and legs black, except for orange trochanters of all legs; abdominal ventrites black with LO in V6 and V7 white, and very narrow lateral and posterior margin of V7 without LO pale cream; abdominal tergites 2–6 black with laterally explanate margins black; T7,8 pale yellowish semitransparent with underlying fat bodies visible; dorsally reflexed margins of V7 pale yellowish.

Pronotum lateral margins divergent (C > A, B); width subequal to humeral width. Elytron: subparallel-sided with interstitial lines evanescent. Head: GHW/SIW 4; ASD subequal to ASW. Abdomen (Figure 6d): LO in V7 filling all of V7, except for narrow, semitransparent lateral and posterior margin; barely distinguishable MPP present, very short, narrow and apically rounded; T8 with posterior margin truncate, sides slightly divergent and paired arms narrowly prolonged. Aedeagal sheath (Figure 6j,k): sternite terminating in paired rounded hairy lobes. Aedeagus (Figure 6g–i): short, squat, L/W 2.2; LL short, slightly inturned at their apices; BP wide, well sclerotised, halves contiguous in anterior midline.

**Female**. Length 15 mm. Colour (Figure 6e,f): Coloured as for male, except for red trochanters 2 and white semitransparent V7,8 (underlying fat bodies confuses interpretation of colour), and narrow black transverse marking anterior to median emargination in V7. Abdomen (Figure 6f): median V7 emargination deep, extending almost to posterior margin of V6, with slight, black, transverse ridge in median area anterior to emargination. Reproductive system not dissected.

**Larva**. Elongate slender, widest across prothorax, tapering towards posterior end. Colour (Figure 6a,b): very dark brown, thoracic terga shiny; all of dorsal surface evenly speckled with small orange circles representing the base area of hairs or spines; median line not paler than rest; ventral surface diffuse grey brown, of prothorax beneath head very dark brown; coxa of all legs dark brown, remainder of legs and paired light organ patches in abdominal V8 pale cream; protergum with paired anterolateral orange brown ovoid areas. All terga, except for the terminal tergum with paired elongated apically pointed projections, and minute spines along edges; protergum with two pairs of projections, one just before posterior corner, the second pair at posterior corner; meso and meta terga with a pair of mid-lateral projections, and a second pair at posterior corners, a third pair along the posterior margin to either side of the median line; all abdominal terga, except for the terminal one with a series of four projections along the posterior margin, one pair at the corners and the second to either side of the median line.

**Figure 6 insects-16-00394-f006:**
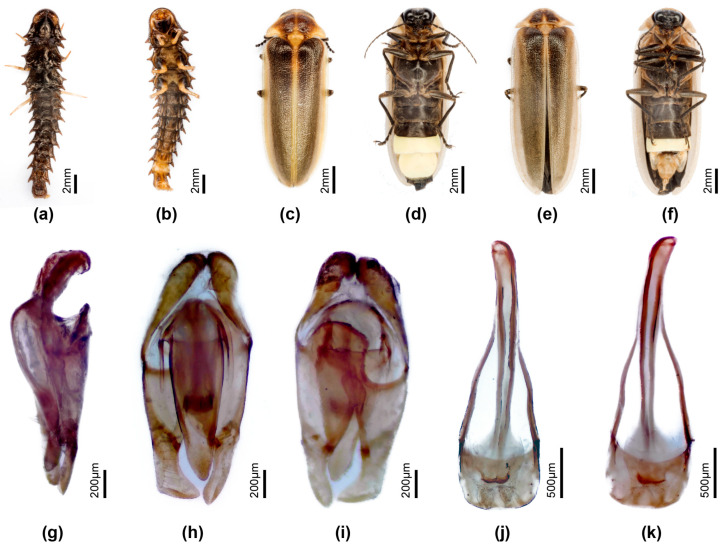
*Pygoluciola yingjiangia* sp. nov. Larva (**a**,**b**), male (**c**,**d**,**g**–**k**), female (**e**,**f**). (**a**,**c**,**e**) Dorsal habitus; (**b**,**d**,**f**) Ventral habitus. (**g**–**i**) Aedeagus: (**g**) right lateral; (**h**) ventral; (**i**) dorsal. (**j**,**k**) Aedeagal sheath: (**j**) ventral; (**k**) dorsal.


**
*Pygoluciola yupingia *
**
**sp. nov.**



Figure 7


**Types**: Holotype. Male. **China**. Collector is Fu, X. H. Sichuan Province, Meishan City, Hongya County, Mt. Yuping. 16.vii. 2023. Paratypes: two males, same data as for holotype (HZMAU).

**Etymology**: The anglicised name for the type locality, Mt. Yuping in Sichuan Province, is latinised and considered a noun in apposition.

**Diagnosis**: Dorsal colouration very similar to that of *P. davidis* comb. nov., *P. qingyu*, and *P. quzhou* sp. nov. Distinguished from *P. davidis* comb. nov. by the black non-margined elytra (elytra of *davidis* with wide pale margins), from *qingyu* and *quzhou* by the vertically expanded paired T8 arms, the inturned LL apices, and the distinctive shape of the median dark pronotal markings (in the latter two species, the arms of T8 are not expanded, LL apices not inturned, and the dark pronotal markings are ovoid).

**Male**. Length 12–14 mm. Colour (Figure 7a–d): Pronotum pale whitish, semitransparent, with pink underlying fat body; fat body retracted along anterior margin and black head visible beneath; paired brown markings on disc not extending to anterior or posterior margins, narrowly separated in the median line, lateral margins divergent, and posterolateral corners slightly produced and rounded; MN slightly pink, MS white; elytra black; lateral edge of prothoracic cavity narrowly black; head, palpi dark brown; antennae with scape, pedicel black, FS 1–3 mid-brown, remainder of FS white; venter of pro and mesothorax whitish cream; metepipleural plates pale cream, metaventrite very dark brown with a very narrow paler posterior margin and scarcely visible median line; legs 1 with whitish cream coxae, basal 1/3 femora, black tibiae and tarsi; legs 2 with whitish cream coxae, trochanters, and almost all of femora and tibiae but for apical brown area on both; legs whitish cream, except for black tarsi and narrow dark brown marking on inner base of coxae; abdomen with basal ventrites dark brown; dorsal abdomen pale brown semitransparent with fat bodies visible, T2-5 and lateral reflexed plates mid-brown; T6 and dorsally reflexed lateral margins whitish semitransparent underlying pale pink fat bodies confuse interpretation of colour; T7 semitransparent with two blocks of pink fat body in anterior half, without fat body in posterior half; T8 greyish cream, semitransparent, no underlying fat body.

Pronotum with lateral margins divergent, (C > A, B); width slightly less than humeral width. Elytron without obvious interstitial lines. Head: GHW/SIW 2.3, ASD > ASW. Abdomen (Figure 7c,d): posterior margin of V6 broadly and shallowly emarginated; LO in V7 occupying all but a very fine lateral and posterior margin; MPP barely distinguishable; T7 (Figure 7j) with lateral areas depressed; T8 (Figure 7k) with slight convergence of lateral margins in posterior 0.25, posterior margin slightly sinuate; anterior arms subparallel-sided along most of their length, expanded in dorsoventral plane. Aedeagal sheath (Figure 7h,i): terminated by paired hairy lobes. Aedeagus (Figure 7e–g): short, broad (L/W 2.3); ML long; LL short, apices inturned; BP wide, well sclerotised, halves contiguous along anterior margin.

Flash pattern. Under mean temperature 21 °C and humidity 85%, flash duration of sedentary flickering males on hanging grass was 375.4 ± 23.4 ms, and flash rate 2.7 ± 0.2 (flashes/s) (n = 16) (Figure 7l). Thus, flash pattern of male *P. yupingia* is much slower than *P. qingyu* (8 flashes/ sec) [8].

Female and larva unknown.

**Figure 7 insects-16-00394-f007:**
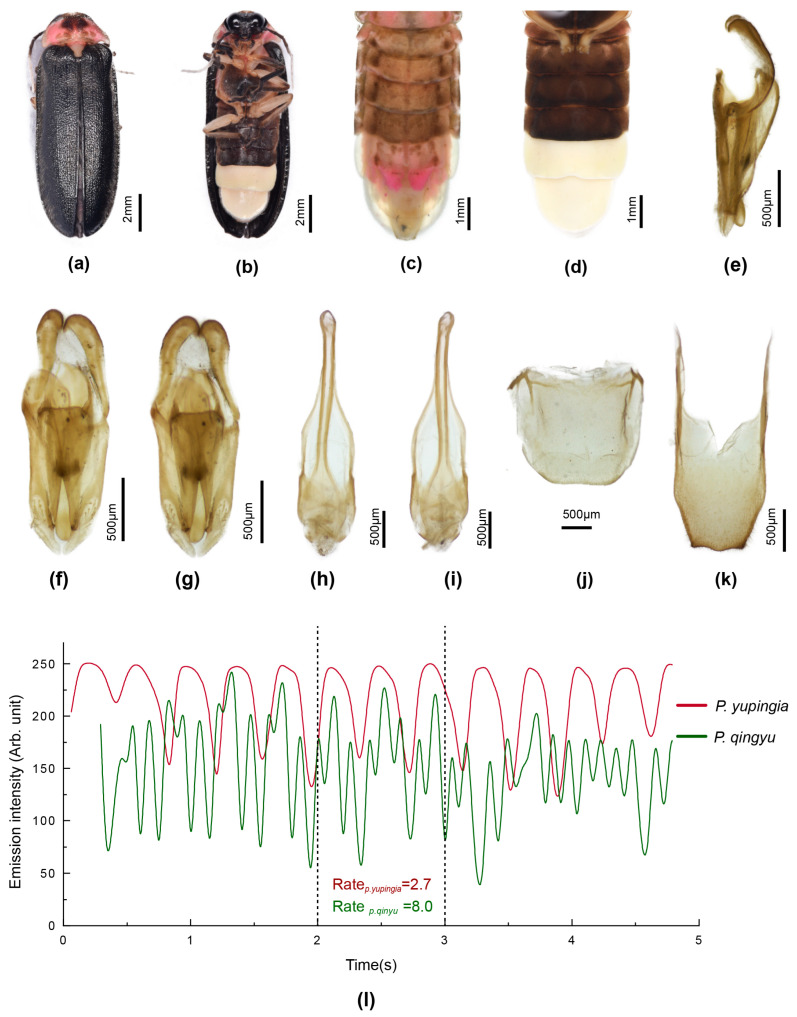
*Pygoluciola yupingia* sp. nov. male. Dorsal habitus (**a**,**c**), Ventral habitus (**b**,**d**). (**e**–**g**) Aedeagus: (**e**) left lateral; (**f**) dorsal; (**g**) ventral. (**h**,**i**) Aedeagal sheath: (**j**) dorsal; (**k**) ventral. (**j**) Tergite 7; (**k**) Tergite 8. (**l**) Comparison of male flash pattern between *P. yupingia* and *P. qingyu*.

**Group 4:** Males: Four small species (ca 6 mm long) from India, Bangladesh, and Myanmar have the rounded posterior V7 margin characteristic of Group 3 but an aedeagus wider across the middle, and aedeagal sheath sternite terminated by a narrow curved section (similar to a boomerang). Body short squat L/W 2/1. Pronotum: 1/6 body length; lateral margins subparallel-sided, (A = B = C), all corners rounded, often obtuse; width less than humeral width. Elytron: interstitial lines not well elevated. Abdomen: posterior margin of V7 either rounded or with an ill-defined short wide horizontal MPP; dorsolateral margins of V7 not investigated, possibly uprolled onto the dorsal surface if dark brown margins surround the LO in V7; T8 subparallel-sided; paired arms not bifurcate at base; posterior margin not narrowed nor downturned, and gently rounded. Aedeagus: short broad, LL short, ML long; widest across middle of aedeagus at junction of basal sclerotised portion of LL and apical membranous section, wider across base and tapering to an acute tip; elongate sausage shaped lobes arising from ventral surface just behind the inner origins of the apical membranous portion of the LL [2] (figure 360). Aedeagal sheath: sternite expanding evenly along length with no defined separate apical expansion; sternite apex almost as wide as tergite.

Female macropterous, pinned, association of two females by label data and similarity of colour pattern in two specimens only to that of males only; external habitus only depicted [2] (figures 394, 395).

Larvae. Unknown.


**
*Pygoluciola abscondita *
**
**(Olivier)**


For taxonomic and figure references to this species, see [2,10,11,41].

**Type**: **Myanmar**. Syntype series of four males (MCSN).

**Diagnosis**: **Male**. A small (6.0–6.2 mm long) species with yellowish orange pronotum often with darker median markings, and brown to dark brown elytra with yellow humeral angle and brown apex; LO in both V6, V7 narrowly retracted along dark brown margins; T8 paler brown than anterior dark brown tergites. Known only from Myanmar.

Females and larvae unknown.


**
*Pygoluciola bangladeshi *
**
**Ballantyne**


For taxonomic and figure references to this species, see [2,11]

**Type**: Male. **Bangladesh** (NMHL).

**Diagnosis**: **Male**. A small species (6.1 mm long) with orange pronotum and brown elytra having brown base, orange yellow lateral and sutural margins and apex; dark brown margins around LO in V7; distinguished from *P. vitalisi* by the brown elytral apex and V7 LO reaching to all margins.

Females and larvae unknown.


**
*Pygoluciola insularis *
**
**(Olivier)**


For taxonomic and figure references to this species, see [2,40,42,43].

**Type**: Female. **India** (MNHN).

**Diagnosis**: **Male**. 7.1–8.0 mm long; orange pronotum, brown elytra with brown apex and basal ¼ orange and wide orange lateral margin; all tergites dark brown; LO in V6,7 not retracted from margins.

Female macropterous, not further investigated. Larva unknown.


**
*Pygoluciola vitalisi *
**
**(Pic)**


For taxonomic and figure references to this species, see [2,10,43].

**Type**: Fragmentary specimen of uncertain sex. **Cambodia** (MNHN).

**Diagnosis**: **Male**. 6.0–6.2 mm long; orange pronotum, dark brown elytra with brown base, narrow orange lateral and sutural margins, apices brown; distinguished from *P. bangladeshi* by the brown elytral apices and the LO in V7 not retracted from the margins.

**Group 5**: Males: Ten species of *Pygoluciola*, including three new species, with aedeagus and aedeagal sheath complex similar to that of Group 1. They are without any prolongations of the posterior margins of either V7 or T8, the posterior margin of T8 is not downturned, and the sheath sternite is not terminated by a boomerang shaped piece, except in *P. nitescens*.

Body outline variable, some elongate slender, others with short squat outline. Pronotum with lateral margins divergent (C > A, B), except for *calceata* (B > A, C) and *cowleyi*, where A = B = C; pronotal width subequal to, slightly less than (in *calceata*, *cowleyi*, *matalangao*) or slightly greater than humeral width (in *nitescens*, *rammale*, *ruhuna*); all corners rounded obtuse (except angulate acute in *matalangao* and 90° in *cowleyi*). Elytral interstitial lines not well defined, except lines 1, 2 in *cowleyi*, *matalangao*. Abdomen: posterior margin V7 either broadly rounded or with a short narrow small MPP (except in *calceata*, where MPP short, wide, apically rounded); LO in V7 narrowly retracted along lateral and posterior margins, except extending to lateral margins in both *nitescens* and *matalangao*, narrowly retracted across posterior margin only in *nitescens*, extending into the MPP of *matalangao*; retracted from all margins including anterior in *cowleyi*; posterior margin emarginate in *baise.* Margins of V7 enfold tergite 8 at the sides in *matalangao* only. T8 subparallel-sided; paired arms expanded in vertical plane and with small triangular ventrobasal projection in *biase*, *manmai*; posterior margin truncate or gently rounded, except in *calceata*, where lateral margins arcuate and midposterior margin shallowly emarginated. Aedeagal sheath: sternite elongate, slender subparallel-sided before expanding to enlarged apex, except for *P. rammale*, with lateral margins expanding evenly along its length, apex truncated and medianly emarginate, or wide boomerang shaped apex in *nitescens*, or lateral margins arcuate, widest in midsection with truncated apex in *ruhuna*. Aedeagus: LL long, ML short, except for *nitescens* and *ruhuna*, where LL are expanded in horizontal plane.


**
*Pygoluciola ambita *
**
**(Olivier)**


For taxonomic and figure references to this species, see [2,10,11,44,45].

**Type**: Male. **Indonesia** (MNHN).

**Diagnosis**: **Male**. 11.0–12.0 mm long. With orange pronotum and brown elytra with orange yellow lateral and sutural margins and apex; LL long, ML short.

Females and larvae unknown.


**
*Pygoluciola baise *
**
**sp. nov.**



Figure 8


**Types**: Holotype. Male. **China**. Collector is Fu, X. H. Guangxi Province, Baise City, 3.vi. 2024. Paratypes: three males, one female, same data as for holotype (HZMAU).

**Etymology**: The specific name is considered a noun in apposition and is retained in its original anglicised form, as this is the name of the City Baise where these specimens were collected.

**Diagnosis**: **Male**. Dorsally brownish yellow with black elytral apices; irregular pale brownish markings in the median area of the pronotum are ill-defined; one of three similarly coloured species in Group 5 having yellowish dorsum with black elytral apices, most obviously distinguished from *P. manmaia* sp. nov. and *P. tunchangia* sp. nov. by the emarginated posterior margin of the V7 LO. Females macropterous, coloured as for male, bursa plates not detected.

**Male**. Length 8–10 mm. Colour (Figure 8a–d): Dorsal surface of pronotum yellowish brown, with irregular ill-defined light brown markings in median disc; underlying fat bodies visible; MS, MN, and elytra yellowish brown, suture slightly paler than rest (probably due to clustering of underlying fat body); head, antennae, and palpi black; ventral aspect of thorax and legs dingy light brown, except for diffuse median dark markings on metaventrite, black tibiae, tarsi, and black inner basal portion of coxae 3; abdominal ventrites black, except for white LO in V6,7; margins of V7 LO opaque, fat bodies clustered narrowly along lateral margin, across posterior margin and into MPP; basal abdominal tergites and reflexed margins of segments 2–5 mid-brown, T7,8 pale yellow semitransparent with narrow light brown posterior margins of T7,8.

Pronotum lateral margins divergent (C > A, B); width subequal to humeral width. Head: GHW/SIW 4; ASD subequal to ASW; head slightly wider than width across prothoracic cavity. Abdomen (Figure 8b–d,n,o): LO in V7 retracted along lateral and posterior areas with lateral margins straight, tapering slightly posteriorly; apically rounded short MPP; T7 (Figure 8n); T8 (Figure 8o) lateral margins subparallel-sided, anterior arms expanded in vertical plane, posterior margin broadly rounded in dissected preparation. Aedeagal sheath (Figure 8l,m): sternite bearing two hairy lobes at apex. Aedeagus (Figure 8i–k): BP narrow, halves almost contiguous along anterior margin; LL long, ML short with apical 1/5 narrow.

**Female**. Length 9–10 mm. Macropterous. Coloured (Figure 8e–h) as for male, except for orange V7, V8, and dark brown T7, T8; fat bodies visible in V7. Abdomen (Figure 8f–h,r,s): median emargination of V7 deep, extending under posterior margin of V6; posterior corners of V7 narrowly acute; V8 apodeme continuous with posterior section; T8 anterior corners narrow acute. Reproductive system (Figure 8p,q) no obvious bursa plates detected but absence not assumed; FAG present.

**Larva** unknown.

**Discussion**. For an exploration of our justification that this and the two similarly coloured species may be regarded as new, see discussion following *P. tunchangia* sp. nov.

**Figure 8 insects-16-00394-f008:**
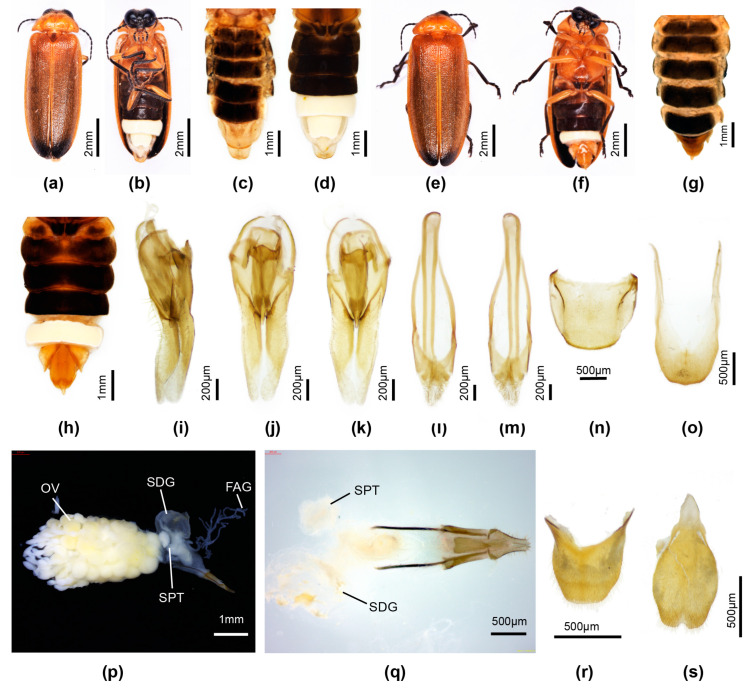
*Pygoluciola baise* sp nov. male (**a**–**d**,**i**–**o**), female (**e**–**h**,**p**–**s**). (**a**,**c**,**e**,**g**) Dorsal habitus; (**b**,**d**,**f**,**h**), Ventral habitus. (**i**–**m**) Aedeagus: (**i**) right lateral; (**j**) dorsal; (**k**)ventral. (**l**,**m**) Aedeagal sheath: (**l**) dorsal; (**m**) ventral. (**n**) Tergite 7; (**o**) Tergite 8. (**p**–**s**) Female reproductive system: (**p**) intact reproductive system; (**q**) ovipositor with bursa, SDG, and SPT; (**r**) tergite 8; (**s**) ventrite 8. Legend: FAG, female accessory gland; OV, ovaries; SDG, spermatophore digesting gland; SPT, spermatheca.


**
*Pygoluciola calceata *
**
**(Olivier)**


For taxonomic and figure references to this species, see [2,10,11,46,47].

**Type**: Male **India** (MNHN).

**Diagnosis**: Male. Striking mid-sized species (9.0 mm long) with unmarked orange pronotum and black elytra; LL long, ML short; T8 with distinctive lateral margins [2] (figure 387).

Females and larva unknown.


**
*Pygoluciola cowleyi *
**
**(Blackburn)**


For taxonomic and figure references to this species, see [10,35,40,48,49,50,51,52,53,54].

**Type**: Male. *Luciola Cowleyi* Blackburn Australia, North Queensland (NHML).

**Type**: Male. *Luciola quadricostata* Pic North Australia (BASEL).

**Diagnosis**: **Male.** Small (ca 5 mm long), yellowish pronotum with wide median brown marking; elytra pale brown with paler lateral margins and interstitial lines 1, 2 well defined; large head, eyes with strong posterolateral excavation.

Female unknown; possible larval association by label data only [55].


**
*Pygoluciola manmaia *
**
**sp. nov.**



Figure 9


**Types**: Holotype. Male. **China.** Collector is Fu, X. H. Yunan Province, Mengla County, 17.iii. 2022. Paratypes: four males, one female, same data as for holotype (HZMAU)**.**

**Etymology**: The type locality of Manmai in Yunnan Province is anglicised and latinised and regarded as a noun in apposition.

**Diagnosis**: Male. Dorsally brownish yellow with black elytral apices; ill-defined irregular pale brownish markings in the median area of the pronotum; one of three similarly coloured species in Group 5 having yellowish dorsum with black elytral apices, males most obviously distinguished from *P. baise* sp. nov. by the non-emarginated posterior margin of the V7 LO (that of *P. baise* sp. nov. is medially emarginated); similar to *P. tunchangia* sp. nov., distinguished by the yellow basal abdominal ventrites (those of *P. tunchangia* are black).

Female macropterous, venter yellow, except for black head, antennae, palpi, tibiae, and tarsi and inner area of coxae 3; distinguished by the broad anterior corners of T7.

Larvae distinguished from *P. tunchangia* sp. nov. by the smaller separated paler markings along the protergum anterior margin (continuous in *tunchangia*), the absence of the tiny speckled orange markings on the dorsal surface of all terga, the narrowly pale brown median line (dark brown in *P*. *tunchangia* sp. nov.), the pale cream coxae with black basal area (*P*. *tunchangia* sp. nov. coxae are black), the projections at the posterior corners of abdominal segments 6–8 longer and wider than the inner two (*P*. *tunchangia* sp. nov. has all terga, except for abdominal tergum 9, with the projections at the posterior corners longer and wider than the inner 2).

**Male.** Length 8–10 mm. Colour (Figure 9c,d): dorsal surface orange yellow with black elytral apices; pronotum with fat body irregularly retracted beneath surface makes interpretation of a possible darker colour pattern difficult; MS slightly paler than rest in anterior half; posterior 1/3 suture paler yellow than rest of elytron, probably due to accumulation of fat body; head, antennae, palpi, all tibiae and tarsi, basal abdominal ventrites, and inner area of coxae 3 black; metaventrite with narrow paired longitudinal brown markings; LO in V6,7 white, lateral and posterior margins V7 semitransparent appearing white laterally due to fat body accumulation; T2-5 and laterally reflexed margins mid-brown; T6-8 and lateral margins of T6 white semitransparent with underlying white fat body

Pronotum lateral margins divergent (C > A, B); width subequal to humeral width. Elytron: faintly defined interstitial lines. Head: GHW/SIW 5; ASD subequal to ASW; head wider than prothoracic cavity and incapable of retraction within the cavity. Abdomen (Figure 9d): LO in V7 filling V7, except for narrow lateral margins and a wider posterior margin; with a well defined apically rounded MPP; T8 (Figure 9q) posterior margin rounded, sides subparallel-sided, anterolateral arms expanded in vertical plane with small triangular projection along outer margin at base of arms. Aedeagal sheath (Figure 9o,p): sternite bearing paired lobes at its apex. Aedeagus (Figure 9l–n): BP narrow, halves not contiguous; LL long; ML short, apex ML narrowed.

**Female.** Length 9–10 mm. Colour (Figure 9e,f): semitransparency of elytra allows underlying dark hind wing to show and confuse interpretation of colour; as for male, with these exceptions: basal abdominal ventrites and V7,8 orange yellow, both more heavily sclerotised than the more anterior segments; LO in V6 only. Abdomen (Figure 9f,t,u): posterior margin of V7 deeply emarginated, not extending beneath posterior margin of V6, posterior corners acute; V8 ovoid, longer than wide, posterior margin shallowly emarginated, anterior prolongation separated from remainder; T8 anterolateral corners widely expanded enfold ovipositor in intact specimen. Reproductive system (Figure 9r,s): well developed FAG; no bursa plates detected but absence not assumed.

**Larva** (Figure 9a,b,g–k). Elongate slender with dorsal plates well sclerotised, some laterotergites visible from above depending on orientation of the specimen. Colour: dorsal surface very dark brown with paired separated ovoid pale yellow markings along anterior margin of protergum; narrow mid-brown median line running to the posterior margin of abdominal tergum 8; all terga with minute ovoid orange markings evenly distributed across surface; ventral surface mid-brown with diffuse darker brown markings beneath retracted head, along lateral areas of meso and metasterna, at sides of abdominal sterna of segments 1–7, 9 and narrowly along ventral surface of laterosternites and laterotergites in segments 1–7; legs pale yellow cream with base of coxae dark brown; LO visible as two distinct round white lateral patches beneath segment 8. Head: Inner margin of mandibles with paired teeth, anterior tooth larger than posterior; palpi with terminal sense organs.

**Figure 9 insects-16-00394-f009:**
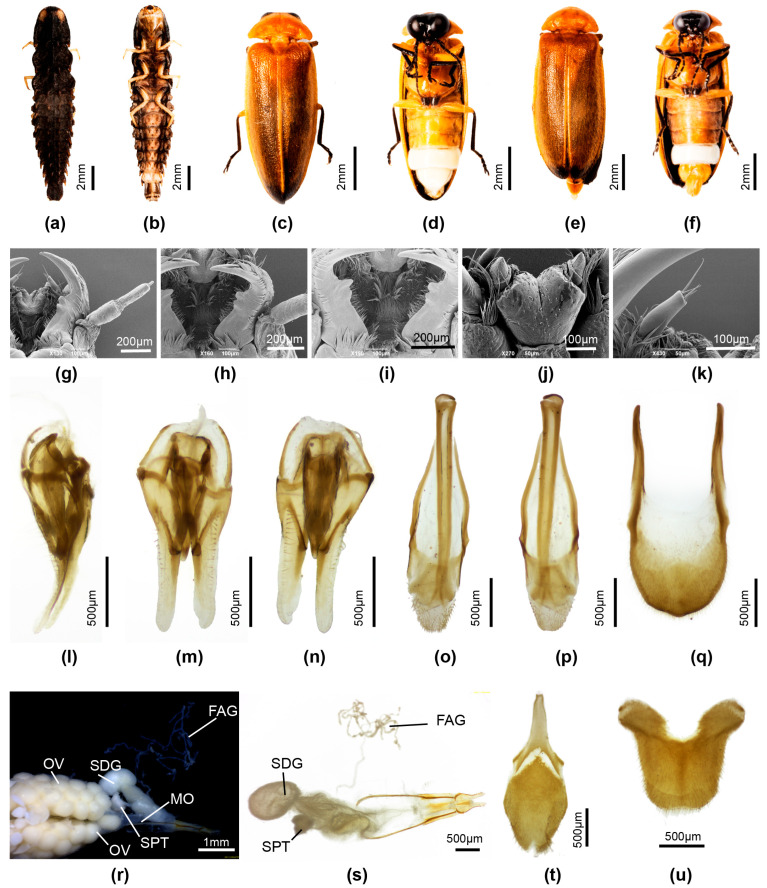
*Pygoluciola manmaia* sp nov. larva (**a**,**b**,**g**–**k**), male (**c**,**d**,**l**–**q**), female (**e**,**f**,**r**–**u**). (**a**,**c**,**e**,**g**–**i**) Dorsal habitus; (**b**,**d**,**f**,**j**,**k**) Ventral habitus. (**l**–**p**) Aedeagus: (**l**) right lateral; (**m**) dorsal; (**n**) ventral. (**o**,**p**) Aedeagal sheath: (**o**) dorsal; (**p**) ventral. (**q**) Tergite 8. (**r**–**u**) Female reproductive system: (**r**) intact reproductive system; (**s**) ovipositor with bursa, SDG, SPT, and FAG; (**t**) ventrite 8; (**u**) tergite 8. Legend: FAG, female accessory gland; MO, median oviduct; OV, ovaries; SDG, spermatophore digesting gland; SPT, spermatheca.


**
*Pygoluciola matalangao *
**
**Ballantyne**


For taxonomic and figure references to this species, see [2].

**Type**: Male. **Philippines** (ANIC).

**Diagnosis: Male.** 8.4−9.8 mm long; with dingy orange pronotum having median dark markings, and medium brown elytra with narrowly paler well defined interstitial lines 1, 2 and suture. Dorsally reflexed lateral margins of V7 enfold sides of T8.

Female macropterous, coloured like male; reproductive system not investigated. Larva unknown.


**
*Pygoluciola nitescens *
**
**(Olivier)**


For taxonomic and figure references to this species, see [2,10,46,56].

**Type**: **India**. Not located.

**Diagnosis**: A large heavy bodied species (12.0 mm long) with orange pronotum and black elytra often narrowly orange marked across their bases; distinguished from the similarly coloured *P. calceata* **comb. nov.** by its size (*calceata* is 9 mm long) and the outline of the MPP (in *nitescens* short, apically acute; in *calceata* about as long as wide, apically truncated).

Female macropterous coloured as for male; reproductive system not showing evidence of bursa plates but absence not assumed [2] (figure 436).

Larva unknown.


**
*Pygoluciola rammale *
**
**Wijekoon & De Silva**


For taxonomic and figure references to this species, see [11].

**Types**: Sri Lanka (DZURSL).

**Diagnosis**: One of two species from Sri Lanka with uniformly yellow unmarked dorsal colouration; distinguished from *P. ruhuna* Wijekoon & De Silva by the aedeagal sheath sternite expanding evenly along its length, apically truncate and medianly emarginated (that of *ruhuna* has lateral margins arcuate, with non-emarginated apex); aedeagus of *rammale* with LL not expanded in horizontal plane (that of *ruhuna* has LL expanded in horizontal plane).

Female macropterous, coloured as for male, minute bursa plates observed.

Larvae not associated. 


**
*Pygoluciola ruhuna *
**
**Wijekoon & De Silva**


For taxonomic and figure references to this species, see [11].

**Types**: Sri Lanka (DZURSL).

**Diagnosis**: One of two species from Sri Lanka with uniformly yellow unmarked dorsal colouration, distinguished from *P. rammale* Wijekoon & De Silva by the aedeagal sheath sternite with lateral margins arcuate, and non-emarginated apex (that of *rammale* has the aedeagal sheath sternite expanding evenly along its length, apically truncate and medianly emarginated); aedeagus of *ruhuna* has LL expanded in a horizontal plane (LL of *rammale* aedeagus are not expanded in a horizontal plane).

Female macropterous, coloured as for male, minute bursa plates observed.


**
*Pygoluciola tunchangia *
**
**sp. nov.**



Figure 10


**Types**: Holotype. Male. **China.** Collector is Fu, X. H. Hainan Province, Tunchang County, 22.v. 2016. Paratypes: three males, one female, same data as for holotype (HZMAU).

**Etymology**: The type locality, Tunchang county Hainan Island, as an English word, and Latinised, is considered a noun in apposition.

**Diagnosis**: *Pygoluciola tunchangia* sp. nov.is one of three species in Group 5 with yellowish brown dorsum and black tipped elytra; distinguished from *P. baise* sp. nov. by the non-emarginated posterior margin of the V7 LO (that of *baise* is emarginated); from *P. manmaia* sp. nov. by the convergent lateral margins and truncate posterior margin of the V7 LO (that of *P. manmaia* sp. nov. has all these margins gently rounded.

Female macropterous, with well defined FAG; no bursa plates detected.

Larva elongate slender with dorsal plates well sclerotised, laterotergites not visible from above; thoracic and abdominal terga have four short acute projections along posterior margins of terga to either side of the median line, and at the posterolateral corners; mandibles have a single narrow apically rounded tooth.

**Male**. Length 8–10 mm. Colour (Figure 10c,d): Dorsal surface of pronotum, MS, MN and elytra orange yellow with black tipped elytra; head between eyes, antennae, and palpi, black; ventral surface of thorax and legs orange yellow, except all tibiae and tarsi black, metaventrite with diffuse dusky median markings, and black inner portions of coxae; abdominal ventrites black with LO in V6, V7 white, and narrow lateral and posterior margin of V7 opaque, fat bodies narrowly clustered along lateral margins; abdominal T2-6 and dorsally reflexed margins black; T7,8 pale yellowish semitransparent with underlying fat bodies visible.

Pronotum. Lateral margins divergent (C > A, B); width subequal to humeral width. Elytron: subparallel-sided with interstitial lines evanescent. Head: GHW/SIW 4/1; ASD subequal to ASW. Abdomen (Figure 10d,q): LO in V7 fills all of V7, except for a narrow lateral and posterior margin; barely distinguishable very short, narrow, and apically rounded MPP present; T8 (Figure 10q) posterior margin rounded, sides slightly divergent, and anterolateral arms expanded in vertical plane with small triangular projection along outer margin near base of arms. Aedeagal sheath (Figure 10o,p): sternite bearing paired lobes at its apex. Aedeagus (Figure 10l–n): BP narrow, halves not contiguous along anterior margin; LL long; ML short; ML apex narrowed.

**Female.** Length 9–10 mm. Colour (Figure 10e,f): semitransparency of elytra allows underlying dark hind wing to show and confuse interpretation of colour; as for male, with these exceptions: basal abdominal ventrites dark brown, V5 very dark brown almost black; LO in V6 only; V7,8 orange yellow, fat body clustered beneath V7. Abdomen (Figure 10f,t,u): posterior margin of V7 deeply emarginated not extending beneath posterior margin of V6, posterior corners rounded; V8 ovoid, posterior margin shallowly emarginated, anterior prolongation separated from remainder; T8 anterolateral corners expanded, apically narrowed with obliquely truncated anterior margins, enfold ovipositor in intact specimen. Reproductive system (Figure 10r,s): well defined FAG; bursa plates not detected but absence not assumed.

**Larva** (Figure 10a,b,g–k). Colour: dorsal surface black with paired wide orange markings along anterior margin of protergum separated by a narrow black median line running to the posterior margin of abdominal tergum 8; all terga with irregular orange markings; ventral surface yellowish in membranous areas with darker brown markings beneath retracted head, along lateral areas of meso and metasterna, all coxae, at sides of abdominal sterna of segments 1–7, 9 and narrowly along ventral surface of laterosternites and laterotergites in segments 1–7; legs pale yellow cream; LO visible as two distinct round white lateral patches in segment 8. Elongate, slender dorsal surface well sclerotised and laterotergites not visible from above; all terga, except terminal abdominal tergum with short narrow apically rounded projections along posterior margin at posterior corners and to either side of the median line and narrowly reflexed ventrally and visible at the margins in ventral view; posterior and lateral margins all terga, except terminal tergum with single line of minute short black spines. Head (Figure 10g–k) conventional form with anterior margin of median epipleural plate with wide shallow emargination; antennae with elongate sense cone subequal in length to AS 3; mandibles with single narrow apically rounded tooth; apical maxillary and labial palpomere with terminal sense organs; labial palpi widely separated.

**Figure 10 insects-16-00394-f010:**
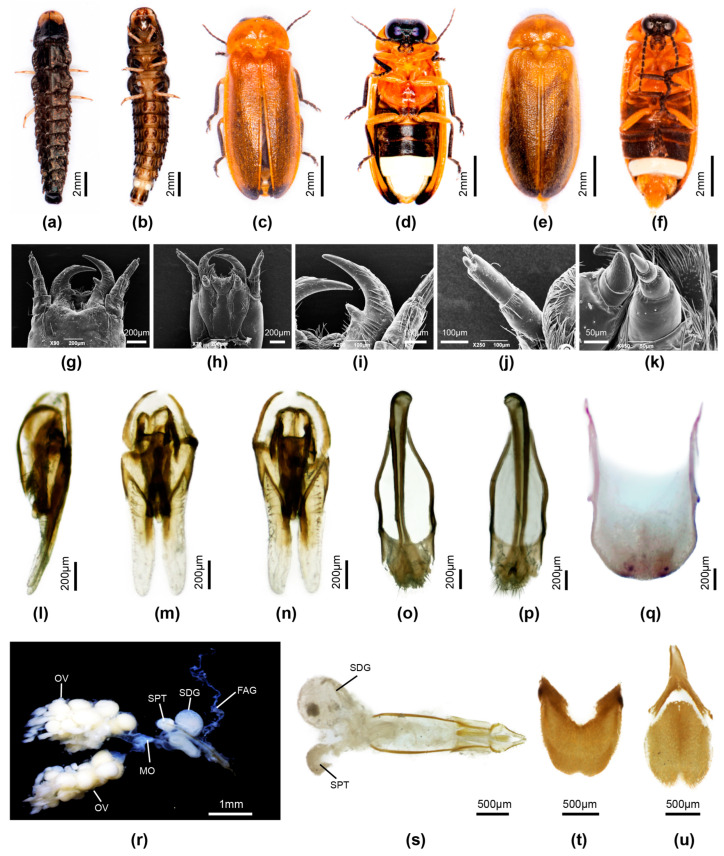
*Pygoluciola tunchangia* sp nov. larva (**a**,**b**,**g**–**k**), male (**c**,**d**,**l**–**q**), female (**e**,**f**,**r**–**u**). (**a**,**c**,**e**,**g**–**i**) Dorsal habitus; (**b**,**d**,**f**,**j**,**k**) Ventral habitus. (**l**–**p**) Aedeagus: (**l**) right lateral; (**m**) dorsal; (**n**) ventral. (**o**,**p**) Aedeagal sheath: (**o**) dorsal; (**p**) ventral. (**q**) Tergite 8. (**r**–**u**) Female reproductive system: (**r**) intact reproductive system; (**s**) ovipositor with bursa, SDG, and SPT; (**t**) tergite 8; (**u**) ventrite 8. Legend: FAG, female accessory gland; MO, median oviduct; OV, ovaries; SDG, spermatophore digesting gland; SPT, spermatheca.

**Remarks**: The dorsal colouration seen in this and two other species in Group 5 is a very common pattern in SE Asia and China [2], and we investigated and attempted to eliminate descriptions of species with similar colouration to determine the accuracy of our contention that this is a new species. Twenty-two species of *Luciola* s. lato, which have similar dorsal colouration to this species, were listed [45] (Tables 6 and 7). These were further addressed in [2], where most were addressed as Species Incertae, as they were impossible to further categorise, being without types or having only female types. Certain species were recommended for transfer to *Curtos*, and revised taxonomic categories were adopted for four. However, several species have features that could suggest an affinity with what we describe here. *L. maculipennis* Olivier [57] from Borneo has no type, but its original description suggested a relationship with *Pygoluciola* [2]. Apart from colouration, no definitive information exists about *L. recticollis* [58] and *L. sordida* [46] from Indonesia, which differ in size (9 mm and 5–6 mm, respectively), with the latter resembling *Colophotia brevis.* In the absence of definitive information suggesting otherwise, we describe this species here as new.

## 4. Discussion

Within the Luciolinae, McDermott (1966) [10] relied largely on the differentiation of genera based largely on obvious (and external) features, including male terminal abdominal modifications. He listed three (*Colophotia* Motschulsky, *Pteroptyx* Olivier, and *Pyrophanes* Olivier), which had terminal abdomen modifications, coupled in *Pteroptyx* with deflexed elytral apices [2]. The lack of obvious differentiating features among many other species led McDermott to assign 279 species to *Luciola* (*Luciola*) Laporte, which, until recently, was the most speciose (and ill-defined) genus in this subfamily; see [2,16]. *Pygoluciola*, with abdominal modifications, was included as a subgenus of *Luciola.*

The taxonomy has developed considerably over the last 16 years due to the discovery of the usefulness of the aedeagus, and especially patterns of the aedeagal sheath (abdominal segments 9 and 10, which are retracted within the abdomen and encase the aedeagus). This has permitted the differentiation of many genera, for which subsequent molecular analyses have provided confirmation [2,11,19]. In particular, an aedeagal pattern, where the LL sit beside the ML, is basic to the initial differentiation in [2] (Appendix A) of 18 genera, including *Pygoluciola*, from those 8 genera where the apices of the LL sit behind the ML.

The significance of the aedeagal sheath in taxonomy was realised in morphological phylogenetic analyses in 2009 [53], and most species of *Pygoluciola* described before then have no aedeagal sheath information. Prior investigative methods often involved pulling the aedeagus out to examine it fully. Aedeagi thus extracted could be left still attached to the abdomen of the pinned specimen. This technique, while revealing parts of the male reproductive system, inadvertently damaged or destroyed the aedeagal sheath. While we make much of aedeagus and sheath patterns in this paper, it is for this reason that we cannot provide descriptions for the patterns in the type species.

The obvious usefulness of the aedeagus and sheath patterns suggested that in *Pygoluciola*, a combination of them with terminal abdomen modifications might be attempted. This first occurred in 2019 [2] when species of *Pygoluciola* were differentiated into three subgroups, subsequently expanded to five in 2024 [11]. These modifications of the aedeagal and aedeagal sheath patterns, together with variations in the developments of the male terminal abdomen, have enhanced our capacity to differentiate new species. We overviewed all species and expanded our definitions of these groups, which has increased their usefulness in taxonomy.

However, while Group 1 is easily distinguished by the more obvious external male terminal abdomen developments, it remains clear that, for some of these species in other groups, confirmation even of genus will still require a preliminary investigation of male terminalia.

This subdivision into five groups is a useful taxonomic exercise, allowing quick identification of males. Two (Groups 3 and 5) are distinguished in our phylogeny (Figure 1). However, while we separated them using terminal abdomen modifications, it is also possible that colour has played a part (the three species in the second clade in Figure 1 are all dorsally yellow with black tipped elytral apices).

Further investigation to determine if other groups may also be supported by molecular information is not presently possible. Species in Groups 1 and 4 have no molecular information (unfortunately, Group 1 houses the type species). Group 2, which comprises only two species, has molecular information of one [16] but was not used in our analyses, which were based on the mitochondrial genome.

While genetic analyses until now have been limited, with their focus usually on *P. qingyu*, they have also used unidentified species (as *Pygoluciola* sp.), for which any morphological confirmation of genus or species has not been possible [13,14,15]. The similarity of genetic patterns has allowed us to suggest possible species identities for two of these species and confirm their placement in the genus (Figure 1).

Collection of fresh material for further investigation is complicated, as many of these species are solitary fliers and are thus difficult to locate and collect [2,3], but the addition of these species constitutes an important next step to illuminate the phylogenetic relationships of all *Pygoluciola* species, their affiliation with the type species, and the evolutionary history of this genus.

Identification of both males and females is further complicated by the very common incidence of the colour pattern of dorsally yellowish with black tipped elytra among several genera and many species in Asia (these include *Abscondita*, *Sclerotia* Ballantyne, and *Triangulara* Pimpasalee among those genera having the aedeagal LL at the sides of the ML [2,45]). Three of our new species (*P. baise* sp. nov., *P. manmaia* sp. nov., and *P. tunchangia* sp. nov.) are so coloured, and we overviewed the 22 species known to possess this colour to justify our assertion these are new species [2,45].

Use of female morphology in identification is more limited, and generic identification of females not taken with a male will almost certainly require an investigation of the female terminal abdomen and reproductive system.

Females can be assigned to generic status, either as *Abscondita* or *Pygoluciola*, based on the hooked bursa plates, However, in several species, these appeared to be either very tiny or possibly absent. We do not yet discount the possibility that additional dissections might reveal bursa plates in some of these species, and given this uncertainty, cannot explain what could be driving their loss (a possible explanation involving abdominal tergites is espoused below). Some species are recognised at the generic level by a combination of the hooked bursa plates and the deeply emarginated posterior margin of V7. Females in Group 1 were dissected only from pinned material. Dissections of fresh material here indicate the presence of an FAG in the female system, which is probably more widespread in the Luciolinae [59].

The female reproductive system conforms to that described for *Pteroptyx* in having a single spermatophore digesting gland and a single stalked spermatheca [60]. The bursa is that section anterior to the entry of the median oviduct housing the paired bursa plates, and the digesting gland arises from its anterior apex.

Bursa plates that are wide and paired (two sets of two fused on each side) are known to hold the spermatophore at least partly protruding into the digesting gland in the female reproductive system of *Pteroptyx maipo* [60] (figure 58). That they could have a similar function in other genera, where there are only two bursa plates, is tacitly assumed but not demonstrated, and such an investigation is beyond the boundaries of this paper. *Aquatica* spp. lay their eggs near bodies of water and have two thin bursa plates, which do not immediately suggest any capacity for holding onto a spermatophore [59]. However, it is interesting that in some species of *Pygoluciola*, very tiny bursa plates have been demonstrated [11], and their possible role in the female system in those species is unclear (but see below). On the other hand, *P. dunguna* has massively developed bursa plates; whether this is an adaptation to the habitat of fresh flowing shallow streams where the females lay their eggs is unknown [7].

Certain features of the female abdominal tergites newly discovered here will prove useful in species determination of the females, but we cannot presently provide comparisons with other species. The discovery that, in *P. kinabalua*, the anterior corners of both T7 and T8 were extended anteriorly beneath the anterior segments suggested increased surface area for muscle attachment (Figure 2). We expanded our investigations in the new species described here. Three species (*P. baise* sp. nov., *P. manmaia* sp. nov., and *P. tunchangia* sp. nov.) all have very distinctively shaped abdominal tergite 8, and in none of these three species were bursa plates detected (Figure 8r, Figure 9u and Figure 10t). It is possible that tergite 8, because of the expanded anterior corners, wraps around the female abdomen, helping to consolidate these terminal segments, but its significance to egg laying is unknown, as we know almost nothing of the habitat requirements. Any pressure exerted may, however, replace the need for bursa plates in helping anchor the spermatophore (Figure 2j, Figure 4j, Figure 5k, Figure 8p, Figure 9r and Figure 10r).

The various terminal abdominal modifications in male Luciolinae species have sparked speculation about their functions [61], but these roles remain largely unexplored due to difficulties in observing nocturnal mating behaviours [2]. However, given the reliance on male terminal abdomen modifications in taxonomy, interpretation of these, particularly in the scoring of characters, is aided by an understanding of their possible functions [2,53,55].

Even with the exaggerated forms of terminalia seen in Group 1 in *Pygoluciola*, Ballantyne’s 1987 [61] observations did not consider they were in any way directly involved in the reproductive process. Rather, she indicated the importance of consideration of ventrite 8, which is missing in the Luciolinae, and would, if present, provide a surface area for attachment of longitudinal muscles. She thought many of the obvious terminal abdomen modifications [61] (p. 178) “may not be of copulatory significance in themselves, but appear to relate either directly or indirectly, to internal areas of muscle attachment”. Alternative interpretations suggest that longitudinal muscles flexing the abdomen may require some stabilisation at the end of the abdomen and that this would involve both V7 and T8 [61]. Specifically, it was posited that the reduced light organ area in V7 in *Pygatyphella* [53] (figure 61) allows for muscle attachment, with the elongated V7 serving as a strengthening rod, against which these muscles pull, inevitably arching against T8 [37,53]. It was proposed that similar explanations could apply to the terminal abdomen elongations in *Pygoluciola.*

It may be significant that *Pygoluciola* can exist successfully in two forms—Group 1, with terminal abdomen modifications to both sternites and tergites, and the remainder without (Group 2 is not considered here, as the modifications are only to ventrite 7). When Fu and Ballantyne first encountered what they described as *P. qingyu*, they were overwhelmed by the obvious similarities in the genitalia to more conventional *Pygoluciola* [8]. If Ballantyne [61] was correct in her explanations, then why can similar species exist in these different forms? Has there been a change in the behaviour such that abdominal flexion is not such a major factor? It may be significant that species in Group 1 are island species [62] restricted to the island of Borneo and the Philippines (Appendix A).

In addition to the attempts to explain modifications of the male terminal abdomen [61], the inclination to attribute them to a possible copulation clamp persists. We know that in *Pteroptyx valida* and *P. maipo*, terminal abdomen modifications, along with the male’s deflexed elytral apex, facilitate a copulation clamp. The female abdomen is wedged between the flattened dorsal surface of the median posterior projection of V7 and the deflexed elytral apex [60,63]. It is only when the participants are in the tail-to-tail position that this clamp functions. It may be significant that it involves an abdominal modification coupled with that of another body part, the deflexed elytral apices. Notwithstanding considerable speculation in this and other subfamilies [64,65] where males have terminal abdomen modifications, these two *Pteroptyx* species remain the only example of a copulation clamp definitively established in fireflies. Wing [63] thought that the bursa plates in *Pt. valida* might provide protection against the pressure of such a clamp, but this was discounted [60], and no external manifestation of a clamp was obvious in females [60].

We did not, however, find other modifications in this genus we could interpret as having clasping functions. The curved tibiae in *P. guigliae* and *P. stylifer* occur in both sexes.

It was thus unexpected when the initial examination of the *P. kinabalua* female abdomen revealed structures that might be involved in an attempt by the male to engage the female abdomen [20] (p. 373). Our reexamination of females of *P. kinabalua* has not provided any additional information that would indicate any copulation clamp could exist, and the presence of the median anterior ridge in V7 of female *P. yingjiangia* sp. nov., where the males have no elongations of the posterior margins of either V7 or T8, indicates that another explanation is necessary.

We discount the possibility that modifications in *P. kinabalua* indicate the presence of a copulation clamp in this species.

Larvae are similar to those of *Abscondita*, with several species in both genera having two mandibular teeth. Both of the larvae described here are associated by breeding. *Pygoluciola qingyu* larvae engage in a head to head battle with ants, but similar behaviour has not been observed in other *Pygoluciola* larvae, and it is not clear if the two mandibular teeth confer that added advantage that allows them to be successful in this exercise [8].

Clearly, the genus *Pygoluciola* Wittmer is notable within the Luciolinae for several features. Primarily, it exhibits pronounced and unusual male abdominal modifications in its type species, *P. stylifer*, as well as in other species [37]. However, we now know that what was previously considered a rare genus is anything but [2,35]. *Pygoluciola* is now known to comprise 28 species, with six of them, all from mainland China, being new, and another (*davidis*) transferred from *Luciola*. Both molecular and morphological information characterises these new species, and unusual in the Luciolinae, 17 species are now characterised by information about the females, in some of which the reproductive system is shown to possess an accessory gland that probably contributes to the egg covering. The expanded use of females and larvae here will help contribute towards development of a Luciolinae taxonomy that is not reliant only on males.

*Pygoluciola* has become the most extensively defined genus in the Luciolinae, as we incorporate features of males (28 species), females (17 species), and larvae (5 species), and species are keyed from all available stages. A molecular analysis incorporating mitogenome sequences establishes the position of the genus as a valid subdivision, its close affinity to *Abscondita*, and supports the morphological division into two of the five groups.

Notwithstanding what has been completed here, exciting prospects await. Collection of some of the rarer species for molecular analysis will help improve the phylogeny, not only of the genus, but the wider Luciolinae. More specimens of more and more species will help us address some of the more puzzling features, like why are tibiae curved in both male and female in some species? Are females really losing their bursa plates? How extensive might the abdominal tergite modifications be, and are they correlated with a loss of bursa plates? Can behavioural observations determine how the mating process proceeds and if there is any indication of a copulation clamp? How many species synchronise?

## Figures and Tables

**Table 1 insects-16-00394-t001:** Numbering of abdominal segments.

Direction	Abdominal Segment	Male	Female
ventral	basal	Ventrite (V) 2	V2
	antepenultimate	V5	V6 light organ
	penultimate	V6 light organ	V7
	terminal	V7 light organ	V8
dorsal	antepenultimate	Tergite (T) 6	T6
	penultimate	T7	T7
	terminal	T8	T8

## Data Availability

Data are contained within the article or Appendix A.

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
