# Peer review of "An Overview of the Firefly Genus *Pygoluciola* Wittmer, a Phylogeny of the Luciolinae Using Mitochondrial Genomes, a Description of Six New Species, and an Assessment of a Copulation Clamp in This Genus (Coleoptera: Lampyridae: Luciolinae) [Author-notes fn1-insects-16-00394]"

_insects, 2025, doi:10.3390/insects16040394_

Round 1
Reviewer 1 Report
Comments and Suggestions for Authors
Fu and Ballantyne once again produced a masterpiece that is likely to significantly impact the field by a solid phylogenetic framework and by facilitating species identification through keys and updated diagnoses. What is more, they made a huge effort to incorporate data on females and larvae, which I commend. In fact, I was very happy to review this paper and to learn from them about this interesting genus of Luciolinae.
In my revised manuscript I suggested a few edits here and there that I believe will improve readability. Below is a list of topics that I think would make this paper even nicer, in my opinion.
1) The authors dedicate some attention to whether or not one of the species has a copulation clamp. This is definitely very important to our field. However, I feel like the authors could have further explored the issue both on the introduction and in the discussion. For example, the authors say: "None of the genera (Pygatyphella, Pygoluciola and Triangulara Pimpasalee) having an 1321 elongated V7 show close relationships in either molecular or morphological analyses [2]" but miss the opportunity to speculate the causes and compare it other known cases of abdominal modifications in the Lampyridae. If they search the recent literature, similar stories were found on the other side of the globe. I must disclose that some of that literature was published by my group, and I know it sounds annoying to be told by the reviewer top "read their papers". Which is why this is just a suggestion, if they feel that this could improve their paper.
2) Yet on the topic of copulation clamps, theory predicts that lock-and-key scenarios would allow for greater species co-occurrences. Which is why I really wish they had added more data on the distribution of each species, or even provided distribution maps by region. In addition to potentially contributing to the discussion of the copulation clamps, that would make it immediately useful to the reader interested in species identification, as it would quickly summarize and convey patterns of geographic distribution.
3) Here: "Abdominal segmentation is not always consistently applied in Coleoptera. Number- 1323 ing of ventral segments follows actual numbers, not visible. While V2 may be the first 1324 visible segment in an entire specimen, when the abdomen of fresh specimens is removed 1325 not only is V2 revealed (in the Luciolinae) to be divided into lateral plaques), but anterior 1326 to this in the membrane we often find small paired plaques that could be attributed to V1." The authors drop an amazingly important information, at the very last paragraph, but I had trouble finding the connection with the rest of the paper, and it read to me more like an afterthought. I wish the authors had embedded it better in the document and made the relevance to their paper more clear.
4) This is more of an aesthetic thing. I think the authors could have avoided the ample blank space between the figures of many plates, which would make room for increasing the size of each figure. Check, for example, figure 3, where my argument may be easier to grasp. That enlargement would be most welcomed for critically relevant structures such as the aedeagus and terminalia.
5) The shape of the female ventrite 7 is very important to species identification, but sufficiently illustrated figures of it are lacking for most species. I hope the authors will consider engaging into illustrating that to go along their very rich descriptions. The same could be said of the male lanterns, but hopefully enlarging the size of the pictures with the advice above will take care of it.
6) The authors refrained from further detailing their methods under the fear that the journal will be suspicious of self-citations. However, it is impossible to properly cover the Luciolinae literature without citing Ballantyne's papers. I urge the editors to rethink this policy or run the risk of improperly omitting critical data. Self-citation in taxonomy is the unavoidable outcome of dealing with challenging groups that no one else works with. It should no be penalized. If anything else, this would be an incredible opportunity/gateway for beginners to learn more about the rich literature available on the Luciolinae.
In sum, I am most impressed – but not surprised – with the overall quality of this work, and I am looking forward to seeing it published. Please do not hesitate if you need any further information from me.
Congratulations to you Fu and Ballantyne!

Author Response
Reviewer 1
Thank you for your detailed comments which, together with those of the second reviewer, have enabled us to make considerable improvements to this paper. We feel that it is much improved with the input from you both.
We have accepted and modified, as per your suggestions, all the material within the document that was not specifically addressed by comments. Any changes show up as bright yellow. The responses to the second reviewer’s comments are indicated in blue. Grey highlight is for items we found ourselves that needed correction. The remainder is addressed below:
M1 additional added - These include the swelling and hardening of the dorsal surface of tergite 7, a transverse ridge in the median area of ventrite 7 anterior to the deeply emarginated posterior margin, and the strongly hooked anterolateral extensions of its anterior margin [20] (page 373).
M2 Will leave this statement there until the editor comments further
M3 omitted
M4 changes made in three areas in response to this comment.
Method:
"Maximum likelihood phylogenies were inferred using IQ-TREE [31] under the GTR+R4+F model for 20000 ultrafast [32] bootstraps, approximate Bayes test [33], as well as the Shimodaira–Hasegawa–like approximate likelihood-ratio test [34]" was changed to
"Maximum likelihood phylogenies were inferred using IQ-TREE [31] under the GTR+R4+F model for 20000 ultrafast [32] bootstraps, as well as the Shimodaira–Hasegawa–like approximate likelihood-ratio test [33]. MrBayes 3.2.6 [34] was used to perform a BI analysis. The analyses of each dataset were performed with 4 MCMC chains and run for 20 million generations. Every 1000th generation was sampled as a consensus tree. The type of consensus tree was Halfcompat. The convergence of the independent runs was indicated by a standard deviation of split frequencies <0.01 and an estimated sample size (ESS) > 200. The initial 25% of sampled data were discarded as burn-in data, and the remaining trees were used to represent the values of posterior probability (PP)."
Result:
"The results of the molecular phylogenetic tree constructed using maximum likelihood are consistent with previous studies based on mitogenome and morphological characters [2]." was changed to
"The results of the phylogenetic tree constructed using the maximum likelihood method and the Bayesian method showed complete consistency,which are consistent with previous studies based on mitogenome and morphological characters [2]. The final analysis results obtained by the two methods were summarized and presented in Figure 1. "
Figure 1 legend:
"Figure 1. Phylogeny of 29 species inferred from concatenation analysis of 37-regions (13 PCGs, 2 rRNA genes and 22 tRNA genes) using maximum likelihood, with GTR+R4+F model and 20000 ultrafast bootstrap replications. Blue values near internodes correspond to bootstrap support/Bayesian posterior probabilities/SHaLRT test. Green values near the branch is the branch length (genetic distance of two genes). The red branches represent the genus Pygoluciola, which contains two sister clades, indicated by the red shade (Clade A) and the blue shade respectively (Clade B)." was changed to
"Phylogeny of 28 Luciolinae species inferred from concatenation analysis of 37 mitogenome regions (13 PCGs, 2 rRNA genes and 22 tRNA genes) using maximum likelihood and Bayesian inference analyses. Tribolium castaneum was used as out group. Blue values near internodes correspond to bootstrap support/Bayesian posterior probabilities/SHaLRT test. Green values near the branch is the branch length (genetic distance of two genes). The red branches represent the genus Pygoluciola, which contains two sister clades, indicated by the red shade (Clade A) and the blue shade respectively (Clade B).".
M5 did you mean call or cull here? I interpreted it as call so have added as many examples of figures that are in this document to illustrate these features.
M6 yes added key reference
M7 addressed
M8 where possible key characters are illustrated at least for one of each couplet; in a couple of instances inserting a new reference would have thrown the reference numbering all completely off; each species has its own section subsequently where figure references are given and I felt this would suffice.
M9 figure references added here and elsewhere see M5 above
M10 added mainland China
M11 size range added (I have only seen two specimens)
M12 Indonesian expanded to include Java.
M13 Australian expanded to include Northern Territory around Darwin
M14 unsure if they are absent or just not seen in this preparation; added “but absence not assumed”.
M15 thanks
M16 I actually saw these specimens back in the 1960s when I was just beginning; this character was not on my radar, and of course then we did not have decent camera equipment so everything had to be drawn. Modified to “not determined in original observation”. For a beginning taxonomist the range of characters one uses is sort off dictated by the number of different specimens and the contrast between them.
M17 We hope to take this on board (Pictures enlargement)
M18 (i) This statement appears on page 12 line 511
List of species of Pygoluciola: see Supplementary Table S1
M18 (ii) patterns of distribution
M19 meaning of having it or not
M20 field observations
M21 explore convergent evolution
M22 omitted related to another paper!!
Additional comments separate to those on the manuscript
- The authors dedicate some attention to whether or not one of the species has a copulation clamp. This is definitely very important to our field. However, I feel like the authors could have further explored the issue both on the introduction and in the discussion. For example, the authors say: "None of the genera (Pygatyphella, Pygoluciola and Triangulara Pimpasalee) having an 1321 elongated V7 show close relationships in either molecular or morphological analyses [2]" but miss the opportunity to speculate the causes and compare it other known cases of abdominal modifications in the Lampyridae. If they search the recent literature, similar stories were found on the other side of the globe. I must disclose that some of that literature was published by my group, and I know it sounds annoying to be told by the reviewer top "read their papers". Which is why this is just a suggestion, if they feel that this could improve their paper.
Response: the discussion has been rewritten and includes several mentions of your own papers in the discussion of abdominal modifications and the copulation clamp.
2) Yet on the topic of copulation clamps, theory predicts that lock-and-key scenarios would allow for greater species co-occurrences. Which is why I really wish they had added more data on the distribution of each species, or even provided distribution maps by region. In addition to potentially contributing to the discussion of the copulation clamps, that would make it immediately useful to the reader interested in species identification, as it would quickly summarize and convey patterns of geographic distribution.
Response: distribution addressed on page 12 line 511 as a supplementary table, and in amended discussion
3) Here: "Abdominal segmentation is not always consistently applied in Coleoptera. Number- 1323 ing of ventral segments follows actual numbers, not visible. While V2 may be the first 1324 visible segment in an entire specimen, when the abdomen of fresh specimens is removed 1325 not only is V2 revealed (in the Luciolinae) to be divided into lateral plaques), but anterior 1326 to this in the membrane we often find small paired plaques that could be attributed to V1." The authors drop an amazingly important information, at the very last paragraph, but I had trouble finding the connection with the rest of the paper, and it read to me more like an afterthought. I wish the authors had embedded it better in the document and made the relevance to their paper more clear.
Response: more properly applied to another paper and what we have done here does not really illustrate this feature (abdomens need to be completely removed from the thorax). This section removed.
4) This is more of an aesthetic thing. I think the authors could have avoided the ample blank space between the figures of many plates, which would make room for increasing the size of each figure. Check, for example, figure 3, where my argument may be easier to grasp. That enlargement would be most welcomed for critically relevant structures such as the aedeagus and terminalia.
Response: figures amended
5) The shape of the female ventrite 7 is very important to species identification, but sufficiently illustrated figures of it are lacking for most species. I hope the authors will consider engaging into illustrating that to go along their very rich descriptions. The same could be said of the male lanterns, but hopefully enlarging the size of the pictures with the advice above will take care of it.
Response: There will need to be further examination of female abdomens as this study has revealed previously undiscovered features not investigated before and this is addressed in the discussion. This relates to both V7 and tergite 7 anterior margin. Where possible descriptions were amended to indicate what is known about this feature.
6) The authors refrained from further detailing their methods under the fear that the journal will be suspicious of self-citations. However, it is impossible to properly cover the Luciolinae literature without citing Ballantyne's papers. I urge the editors to rethink this policy or run the risk of improperly omitting critical data. Self-citation in taxonomy is the unavoidable outcome of dealing with challenging groups that no one else works with. It should no be penalized. If anything else, this would be an incredible opportunity/gateway for beginners to learn more about the rich literature available on the Luciolinae.
Response: will wait for opinion from editor. Thank you for the support.
Reviewer 2 Report
Comments and Suggestions for Authors
Review Insects: An overview of the firefly genus Pygoluciola Wittmer, a phylogeny of the Luciolinae using mitochondrial genomes, a description of six new species, and an assessment of a copulation clamp in this genus (Coleoptera: Lampyridae: Luciolinae)
This manuscript adds the description of 6 new species in Pygoluciola, while redefining this important genus for Luciolinae taxonomy. The morphological traits of 28 Pygoluciola species in 5 morphologically defined species groups, and species keys for male and for female traits are presented. In addition, the authors sequenced the mitogenomes (37 loci) of 7 Pygoluciola species in 2 of the 5 species groups, which showed that these two morphologically defined groups indeed represent two different and well-supported clades in a Luciolinae mitogenome phylogeny (11 genera). It also further supports Abscondita as the sister genus of Pygoluciola.
This contribution is likely of great interest to the readers of Insects, especially those with an interest in beetle taxonomy and evolution. The addition of larval (5 species) and female (17 species) traits greatly expand the utility of species keys for the species identification of Pygoluciola, and highlight the importance of females for taxonomy in general. The extensive morphological descriptions, together with the addition of species-specific male flash patterns (for two species) to taxonomic descriptions– this study of Pygoluciola taxonomy is a great example of the extended specimen approach in modern taxonomy.
I do believe that the self-citation restrictions should be relaxed for this study, especially since one of the authors of this study did most of the fundamental taxonomic work in this group.
Before this manuscript can be published several improvements should be made. My suggested edits fall into three main categories:
- The manuscript would benefit from more clarity in writing for a reader, who may not be as familiar as the authors with the context and/or details of their research. Please provide context (and relevance) to guide the reader, and be as specific as possible.
- Correct typos and labels as needed, and make sure to use the correct phylogenetic terminology.
- Reorganize the discussion to move from the findings of this study (details) back to the big picture. Start with morphological findings and end with their integration into the Luciolinae phylogeny.
Specific comments, questions and suggestions
73: “We aim to address several issues of a taxonomic nature enhanced with molecular information: 1. Pygoluciola is redescribed, the five subgroups are evaluated and elaborated further using morphological and molecular information, keys to species using both males and females are presented; 2. Six new species are described using features of males, and in some[how many?] cases females and [how many?] larvae, and newly determined molecular information; 3. Luciola davidis Olivier is redescribed, assigned to Pygoluciola, and distinguished from P. qingyu by morphological and molecular data; 4. Re-examination of modifications [of what?] of the female P. kinabalua allowed us to suggest alternative explanations [for these structures]. By integrating and expanding the molecular information about the Luciolinae, we present comprehensive phylogenetic trees to clarify intrageneric relationships in the Luciolinae. We believe the collaboration of a taxonomic entomologist and a molecular biologist will help overcome existing identification challenges [like what specifics?] addressed above. The expanded use of females and larvae here will contribute towards development of a Luciolinae taxonomy which is not reliant only on males.”
Comment: The paragraph above is a bit too vague (specifics needed) and confusing (order) to give the reader a good overview. It also is a mixture of aims and results. Can this be made more specific (and accurate)?
For example: This study, a collaboration between a taxonomic entomologist and a molecular biologist, helps overcome existing identification challenges, advances Pygoluciola taxonomy and clarifies the phylogenetic relationship of Pygoluciola with other Luciolinae. This is achieved in 5 steps: 1. Pygoluciola is redescribed, the five current morphological subgroups are evaluated, providing revised definitions and diagnoses for species within each group; 2. Six new Pygoluciola species are described using features of males, but also females (? species) and larvae (? species); 3. Luciola davidis Olivier is redescribed, assigned to Pygoluciola, and distinguished from P. qingyu; 4. Keys to all currently recognized 28 Pygoluciola species are presented, for both males and females (where available); 5. Re-examination of the suggested copulation clamp in female P. kinabalua suggests an alternative explanations [Note: What is the relevance of this for taxonomy: put in context of above/big picture? This unclear in the whole manuscript]; 6. The mitogenomes of 7 Pygoluciola species from two different morphological groups are sequenced and integrated in a Luciolinae phylogeny.
136: “Flash pattern analysis. Field conditions of temperature and humidity as well as flash activity in relation to time after sunset were recorded… The flash duration, interval and rate were calculated”
Comment: For how many Pygoluciola taxa are flash patterns known? Can these be added to more species descriptions (e.g. P. qingyu: only mentioned in comparison to P. yupingia)?
197ff Results:
Comment: Results are usually written in present tense.
206: “From the results, all nodes of the phylogenetic tree obtained high posterior probabilities (Bayesian posterior probabilities > 0.955), although the bootstrap support of the Aquatica branch and the (Luciola + (Abscondita + Pygoluciola)) branch was only 28.8.”
Comment: the terminology used here is not quite accurate. A branch represents an evolving (ancestral or present) species that splits into two as speciation occurs.
I suggest rephrasing it as: All nodes of the phylogenetic tree were supported by high posterior probabilities (Bayesian posterior probabilities > 0.955), although the bootstrap support for the node connecting Aquatica and (Luciola + (Abscondita + Pygoluciola)) was only 28.8 (Figure 1).
209: “All the species of the genus Pygoluciola clustered into the same branch. Furthermore, they can be divided into two sister clades. Clade A represented by P. qingyu is larger in volume with black elytra, while Clade B represented by P. baise sp. nov. is smaller in volume with yellow elytra.”
Comment: Terminology.
I suggest rephrasing this as: The eleven Pygoluciola species in our phylogenetic analysis form a well-supported (99.6/1/99) clade, which contains two clearly distinct sub clades: Clade A (100/1/100) and Clade B (100/1/100). Clade A contains species from morphological Group 3 (e.g. P. qingyu a large firefly with black elytra), and Clade B contains species from morphological Group 5 (e.g. P. baise sp. nov., a small firefly with yellow elytra).
213: “Three unknown species of the genus Pygoluciola were published in NCBI. Pygoluciola sp (MZ571356) and P. quzhou sp. nov. clustered into one branch. Pygoluciola sp. (OP747324), collected in Thailand, had the closest genetic distance to P. baise sp. nov. Pygoluciola sp. FM18 (MK292102) and P. manmaia sp. nov. clustered into one branch. The collection site of Pygoluciola sp. FM18 was Mengla, Yunnan Province, China, which was very close to the collection site of the type specimen of P. manmaia sp. nov. The genetic clustering of its mitochondrial genome sequence was only 0.0059. Therefore, we speculate \that Pygoluciola sp. FM18 is P. manmaia sp. nov.”
Comment: Terminology.
I suggest rephrasing this as: We included the mitogenome sequences of three unidentified Pygoluciola species from Genbank (NCBI REF) in our phylogeny (Figure 1). Pygoluciola sp. (NCBI Genbank ID: MZ571356) clustered within Clade A (Group 3), as the sister taxon to P. quzhou sp. nov. Within Clade B (Group 5), Pygoluciola sp. (NCBI Genbank ID OP747324) from Thailand, was the sister taxon to P. baise sp. nov., and Pygoluciola sp. FM18 (NCBI Genbank ID: MK292102) was the sister taxon of P. manmaia sp. nov. However, Pygoluciola sp. FM18 and P. manmaia sp. nov were separated by very short branchlengths (0.0003+0.0056= 0.0059) and the collection site of Pygoluciola sp. FM18 (Mengla, Yunnan Province, China), is very close to the collection site of the type specimen of P. manmaia sp. nov. Therefore, Pygoluciola sp. FM18 may be P. manmaia sp. nov.
222 (Figure 1 legend): “Phylogeny of 29 species inferred from concatenation analysis of 37-regions (13 PCGs, 2 rRNA genes and 22 tRNA genes) using maximum likelihood, with GTR+R4+F model and 20000 ultrafast bootstrap replications. Blue values near internodes correspond to bootstrap support/Bayesian posterior probabilities/SHaLRT test. Green values near the branch is the branch length (genetic distance of two genes). The red branches represent the genus Pygoluciola, which contains two sister clades, indicated by the red shade (Clade A) and the blue shade respectively (Clade B). Green values near the branch is the branch length (genetic distance of two genes).”
Comment: Please rephrase for accuracy
Example: Figure 1. Phylogeny of 29 Luciolinae species inferred from concatenation analysis of 37 mitogenome regions (13 PCGs, 2 rRNA genes and 22 tRNA genes) using maximum likelihood, with GTR+R4+F model and 20000 ultrafast bootstrap replications. Red branches: Pygoluciola, with two sister clades: Clade A (Group 3: red shading) and Clade B (Group 5: blue shading). Blue node values: bootstrap support/Bayesian posterior probabilities/SHaLRT test. Green values: terminal branch lengths for Pygoluciola sp. FM18 and P. manmaia sp. nov.
251: “C> A, B”
Comment: This needs to be defined at first use. Is there a figure with labels?
281: “dunguna [7] (Figures 34–37); bursa plates very reduced in P. rammale, P. ruhuna [11] (Figures 22, 45) “
Comment: Are these figures inside the references? If yes, I suggest including them in the brackets: “dunguna [7: Figures 34–37]; bursa plates very reduced in P. rammale, P. ruhuna [11: Figures 22, 45] “- otherwise, readers interpret this as a reference to your own figures. Adjust all LIT - figure references in the manuscript (or check with editor).
742: “Female. Length 12-14 mm. Distinguished from P. qingyu female by the dark brown 742 ventral colouration (P. quzhou sp. nov. has pale brown V2-5 before the white V6 LO.”
Comment: P. qingyu female on line 742 should read P. quzhou sp. nov female.
752: Figure 4 labels
Comment: Your labels of the figures in Figure 4 (and male/female labels) are out of order: please check and correct.
821: “Male. Length 13-15 mm. Colour (Figure 5 c,d)..”
Comment: Change Fig 5 references to Fig 6 in this whole section.
910: “Thus, flash pattern of male P. yupingia is much slower than P. qingyu [8 flashes/ sec [8]. “
Comment: Please add the flash information for P. qingyu under its description. Also add any other available flash descriptions (if any) to species descriptions.
932: Text/Figure order
Comment: Move Figure 7 (Group 3) above description of Group 4 males.
933: Figure 7. Pygoluciola yupingia sp. nov. male. (a,c) Dorsal habitus; (a,c), ; (b,d) Ventral habitus. (b,d) Aedeagus: (e) left lateral; (f) dorsal; (g) ventral. (h,i) Aedeagal sheath: (j h) dorsal; (k i) ventral. (j) Tergite 7 ; (k) Tergite 8; (I) flash pattern of one male perched in hanging grass.
Comment: Please correct labeling (and make comparable to other Figure legends).
1157: P. rahuna
Comment: Please specify: male, female unknown?
1259: “However, we did not locate hooked bursa plates in all females we examined.”
Comment: please add for the general reader the context/relevance of this trait for taxonomy, function?
1261: “We discount the possibility that modifications in P. kinabalua indicate the presence of a copulation clamp in this species.”
Comment: please add context/relevance of this trait or presence/absence of a copulation clamp in this species for the interested reader (and for taxonomy of group).
1263: “In fact, Pygoluciola becomes the most extensively defined genus in the Luciolinae as we incorporate features of males, females and larvae, and species are keyed from all stages Luciolinae as we incorporate features of males, females and larvae, and species are keyed from all stages.”
Comment: please specify to avoid false impression that they are available for all, for example:
In fact, Pygoluciola becomes the most extensively defined genus in the Luciolinae as we incorporate features of males (28 species), females (17 species) and larvae (5 species), and species are keyed from all available stages.
1267: “The grouping into five morphological groups remains partly a useful taxonomic exercise allowing quick identification and keying to species. However, at least two of these groups are distinguished in our phylogeny, although interestingly while we distinguished them using terminal abdomen modifications it is also clear that colour has played a part (the three species in the second clade in Figure 1 are all dorsally yellow with black tipped elytral apices). We were unable to investigate further; species in Groups 1 and 4 have no molecular information (and unfortunately Group 1 houses the type species). Collection of fresh material is further complicated as many of these species have been shown to be solitary fliers [2, 3].”
Comment: I feel you are too negative (and vague) about your results here. Please rephrase as findings and next steps, rather than as shortcomings of this study. Question: If there is molecular info for group 2 (“none for 1 and 4”): why was it not included in this study?
Example: The grouping of Pygoluciola into five morphological groups is a useful taxonomic exercise allowing quick identification and keying to species. In addition, our analysis shows that at least two of these groups (group 3 and group 5) with their terminal abdominal modifications (and possibly their coloration) are phylogenetically distinct clades. Whether this also applies to the other 3 groups (groups 1, 2 and 4) remains to be tested. This is especially important for group 1, which contains the type species for the genus. Many of the species [in group 1? others?] are solitary fliers [2, 3], complicating the collection of fresh material for a molecular analysis, but the addition of these species constitutes an important next step to illuminate the phylogenetic relationships of all Pygoluciola species, their affiliation with the type species and the evolutionary history of this genus.
1276 ff: Several paragraphs on detailed morphology until end of manuscript.
Comment: A lot of morphological information (and shortcomings of methods) are listed, but it is unclear why. Each new topic needs an introduction (context). Please give context and logically connect the different paragraphs: right now they read like a collection of disconnected bulletpoints. Only traits are listed, not what they mean or their relevance for taxonomy.
What are the question to be answered? Utility of specific characters for taxonomy? Support for findings of this study? How does it all go together?
For example why is aedagus/sheet important? How does this new info relate to the Pygoluciola phylogeny? Can you diagnose Group 3 and 5 by aedagus/sheet? Make connection before you go to Luciolinae (which?) and how this relates to your study. I am sure this is clear for the expert minds of the authors, but not to the reader of Insects who may not be as immersed in the morphology of this group. Also please note: in the Discussion you usually start with your findings/details and expand back out to big picture (Literature), not the other way around.
Question: are you discussing the use of morphological traits that you identified as defining traits for the 5 Pygoluciola groups as potentially useful for other species in subfamily? And/or to make them more universally useful: how should specimens be treated, traits scored? (suggested improvements for next studies?)
1276: Luciolinae (other than/including Pygoluciola?)- does trait distinguish groups 3 and 5? – also shown for other Luciolinae? strong character for whole subfamily!?? -> future studies: traits need to be used? Be specific.
1279: “The significance of the aedeagal sheath in taxonomy was not realised until [53]”
Comment: please give at least year, if not name. Don’t force reader to scroll through references.
1279: aedagus/sheath section – this section reads defensive!
Why not reframe as: your study supports the significance of aedagus/sheet for Pygoluciola taxonomy. Difficulty with pulling aedagus/ -> what should taxonomist do in the future to fully get data? Aedagal sheet often destroyed and therefore data not available for most specimen, including type species. What is your suggestion? What should the methods be for future studies?
1289: female morphology– [differentiation for what?], hooked bursa plates- what can you do with it/not?. Please be specific so reader can follow. Is this about limitations of morphological analysis? What is the point of this paragraph? Please clarify.
1294: terminal modification of male Luciolinae – what traits are these? start with your insights on Pygoluciola then discuss Luciolinae? Note: in Discussion you usually start with your findings/details and expand back out to big picture (Literature).
1301: context: first introduce copulation clamp and why absence/presence is key here for taxonomy (big picture) rather than just focusing on interpretation of specific structures. I am still not sure about this element after reading the manuscript.
Similarly, clarify the argument you are making with the information in paragraphs 1312, 1317, 1321.
Final comment: The morphological details (1276ff) should come earlier in the discussion. By ending with all the morphological details (and reasons why some traits are not useable, methods were not good, etc.) the reader is left with a negative impression and a bit lost (not sure what to do with all that information). This distracts from the contribution your study is making. Your discussion should end at the big picture/major insights. Discuss morphology first and what you learned from it about Pygoluciola relationships, which traits are key (and how future studies should collect/use them), then end with how morphological groups relate to the phylogeny and the phylogenetic relationships of Pygoluciola (integrate findings) and the other Luciolinae (literature) in this study. Which next study should be done (next question)? This way this manuscript ends on a high note with a clear take-home message.
Author Response
Reviewer 2 Response
Thank you for your extensive and helpful review. We hope we have done it justice and that you are pleased with what we now present, as we feel that with your help it is a much improved paper.
We have accepted and modified , as per your suggestions, all the material within the document that was not specifically addressed by comments. The responses to the first reviewer’s comments are indicated in bright yellow. The responses to the second reviewer’s comments are indicated in blue. Grey highlight is for items we found ourselves that needed correction. The responses are addressed below:
Top of Form
- The manuscript would benefit from more clarity in writing for a reader, who may not be as familiar as the authors with the context and/or details of their research. Please provide context (and relevance) to guide the reader, and be as specific as possible.
Response: hopefully we have achieved that especially in the newly written discussion.
- Correct typos and labels as needed, and make sure to use the correct phylogenetic terminology.
Response: We accepted all revisions made by reviewer regrading of typos, labels and phylogenetic terminology.
- Reorganize the discussion to move from the findings of this study (details) back to the big picture. Start with morphological findings and end with their integration into the Luciolinae phylogeny.
Response see 1 above.
Specific comments, questions and suggestions
73: “We aim to address several issues of a taxonomic nature enhanced with molecular information: 1. Pygoluciola is redescribed, the five subgroups are evaluated and elaborated further using morphological and molecular information, keys to species using both males and females are presented; 2. Six new species are described using features of males, and in some[how many?] cases females and [how many?] larvae, and newly determined molecular information; 3. Luciola davidis Olivier is redescribed, assigned to Pygoluciola, and distinguished from P. qingyu by morphological and molecular data; 4. Re-examination of modifications [of what?] of the female P. kinabalua allowed us to suggest alternative explanations [for these structures]. By integrating and expanding the molecular information about the Luciolinae, we present comprehensive phylogenetic trees to clarify intrageneric relationships in the Luciolinae. We believe the collaboration of a taxonomic entomologist and a molecular biologist will help overcome existing identification challenges [like what specifics?] addressed above. The expanded use of females and larvae here will contribute towards development of a Luciolinae taxonomy which is not reliant only on males.”
Response paragraph modified to
This study, a collaboration between a taxonomic entomologist and a molecular biologist, helps overcome existing identification challenges, advances Pygoluciola taxonomy and clarifies the phylogenetic relationship of Pygoluciola with other Luciolinae. This is achieved in 5 steps: 1. Pygoluciola is redescribed, the five current morphological subgroups are evaluated, providing revised definitions and diagnoses for species within each group; 2. Six new Pygoluciola species are described using features of males, but also females (? species) and larvae (? species); 3. Luciola davidis Olivier is redescribed, assigned to Pygoluciola, and distinguished from P. qingyu; 4. Keys to all 27 currently recognized Pygoluciola species are presented, for both males and females (where available); 5. The possibility that the male terminal abdomen modifications in some species might function as a copulation clamp is investigated not from males but the female abdomen of P. kinabalua suggests an alternative explanations [Note: What is the relevance of this for taxonomy: put in context of above/big picture? This unclear in the whole manuscript]; 6. The mitogenomes of 7 Pygoluciola species from two different morphological groups are sequenced and integrated in a Luciolinae phylogeny.
136: “Flash pattern analysis
Response: We only have flash patterns of P. yupingia and P. qingyu. We compared male flash patterns between P. yupingia and P. qingyu and incorporated them in Figure 7l.
Unfortunately, we don’t have flash patterns of other Pygoluciola fireflies. We will try to record flash patterns of them in further field investigations. .
206: Response: Modified as suggested:
From the results, all nodes of the phylogenetic tree obtained high posterior probabilities (Bayesian posterior probabilities > 0.955), although the bootstrap support of the node connecting Aquatica + Nipponoluciola branch and the (Luciola + (Abscondita + Pygoluciola)) branch was only 28.8 (Figure 1).
209: Terminology.
Response modified as suggested: The eleven Pygoluciola species in our phylogenetic analysis form a well-supported (99.6/1/99) clade, which contains two clearly distinct sub clades: Clade A (100/1/100) and Clade B (100/1/100). Clade A contains species from morphological Group 3 (e.g. P. qingyu a large firefly with black elytra), and Clade B contains species from morphological Group 5 (e.g. P. baise sp. nov., a small firefly with yellow elytra).
213: Terminology.
Response modified as suggested (slight modifications): We included the mitogenome sequences of three unidentified Pygoluciola species from Genbank (NCBI REF) in our phylogeny (Figure 1). Pygoluciola sp. (NCBI Genbank ID: MZ571356) clustered within Clade A (Group 3), as the sister taxon to P. quzhou sp. nov. Within Clade B (Group 5), Pygoluciola sp. (NCBI Genbank ID OP747324) from Thailand, was the sister taxon to P. baise sp. nov., and Pygoluciola sp. FM18 (NCBI Genbank ID: MK292102) was the sister taxon of P. manmaia sp. nov. However, Pygoluciola sp. FM18 and P. manmaia sp. nov were separated by very short branch lengths (0.0003+0.0056= 0.0059) and the collection site of Pygoluciola sp. FM18 (Mengla, Yunnan Province, China), is very close to the collection site of the type specimen of P. manmaia sp. nov. Therefore, Pygoluciola sp. FM18 may be P. manmaia sp. nov.
222 (Figure 1 legend): Modification includes some of your suggestions: Figure 1. Phylogeny of 29 Luciolinae species inferred from concatenation analysis of 37 mitogenome regions (13 PCGs, 2 rRNA genes and 22 tRNA genes) using maximum likelihood, and Bayesian inference analyses. Tribolium castaneum was used as out group. Blue values near internodes correspond to bootstrap support/Bayesian posterior probabilities/SHaLRT test. Green value near the branch is the branch length (genetic distance of two genes). The red branches represent the genus Pygoluciola, which contains two sister clades, indicated by the red shade (Clade A) and the blue shade respectively (Clade B).
251: “C> A, B”
Response: A B and C added to list of abbreviations at end of paper.
281: Response: two issues here addressed – a. positioning of figure references all checked and adjusted and this added at line 160
To avoid confusion figure references attached to literature are written as “figure” while references to figures within this manuscript are written as “Figure”.
- descriptions of female abdomen modified to include details of V7 and T7, with figures and literature reference added where possible.
Comment: P. qingyu female on line 742 should read P. quzhou sp. nov female.
Response: altered.
Comment: Figure 4 captions amended.
Response: amended.
821: “Male. Length 13-15 mm. Colour (Figure 5 c,d)..”
Comment: Change Fig 5 references to Fig 6 in this whole section.
Response: amended.
910: “Thus, flash pattern of male P. yupingia is much slower than P. qingyu [8 flashes/ sec [8]. “
Comment: Please add the flash information for P. qingyu under its description. Also add any other available flash descriptions (if any) to species descriptions.
Response: added for qingyu; there are only two species for which we have flash patterns
932: Text/Figure order
Comment: Move Figure 7 (Group 3) above description of Group 4 males
933: Figure 7. Comment: Please correct labeling (and make comparable to other Figure legends).
Response: modified..
1157: P. rahuna
Comment: Please specify: male, female unknown?
Response: descriptions of both rammale and ruhuna expanded.
1259: “However, we did not locate hooked bursa plates in all females we examined.”
Comment: please add for the general reader the context/relevance of this trait for taxonomy, function?
Added to methods section and expanded in the discussion
Dissections of bursa plates were performed by different people in different continents and Ballantyne did not see all of these final female dissections. Because of the potential of subjectivity in interpretation of presence or absence of these plates, we have amplified our descriptions where none was seen to indicate also that absence not assumed”. Relevance of bursa plates expanded in discussion
1261: “We discount the possibility that modifications in P. kinabalua indicate the presence of a copulation clamp in this species.”
Comment: please add context/relevance of this trait or presence/absence of a copulation clamp in this species for the interested reader (and for taxonomy of group).
Added to materials section around line 161 and in discussion.
Response: from this point on the comments address the discussion which has been completely rewritten; it is almost impossible to address each comment separately but we have been guided by those in this rewrite.
1263: “In fact, Pygoluciola becomes the most extensively defined genus in the Luciolinae as we incorporate features of males, females and larvae, and species are keyed from all stages Luciolinae as we incorporate features of males, females and larvae, and species are keyed from all stages.”
Comment: please specify to avoid false impression that they are available for all, for example:
In fact, Pygoluciola becomes the most extensively defined genus in the Luciolinae as we incorporate features of males (28 species), females (17 species) and larvae (5 species), and species are keyed from all available stages.
1267: “The grouping into five morphological groups remains partly a useful taxonomic exercise allowing quick identification and keying to species. However, at least two of these groups are distinguished in our phylogeny, although interestingly while we distinguished them using terminal abdomen modifications it is also clear that colour has played a part (the three species in the second clade in Figure 1 are all dorsally yellow with black tipped elytral apices). We were unable to investigate further; species in Groups 1 and 4 have no molecular information (and unfortunately Group 1 houses the type species). Collection of fresh material is further complicated as many of these species have been shown to be solitary fliers [2, 3].”
Comment: I feel you are too negative (and vague) about your results here. Please rephrase as findings and next steps, rather than as shortcomings of this study. Question: If there is molecular info for group 2 (“none for 1 and 4”): why was it not included in this study?
Example: The grouping of Pygoluciola into five morphological groups is a useful taxonomic exercise allowing quick identification and keying to species. In addition, our analysis shows that at least two of these groups (group 3 and group 5) with their terminal abdominal modifications (and possibly their coloration) are phylogenetically distinct clades. Whether this also applies to the other 3 groups (groups 1, 2 and 4) remains to be tested. This is especially important for group 1, which contains the type species for the genus. Many of the species [in group 1? others?] are solitary fliers [2, 3], complicating the collection of fresh material for a molecular analysis, but the addition of these species constitutes an important next step to illuminate the phylogenetic relationships of all Pygoluciola species, their affiliation with the type species and the evolutionary history of this genus.
1276 ff: Several paragraphs on detailed morphology until end of manuscript.
Comment: A lot of morphological information (and shortcomings of methods) are listed, but it is unclear why. Each new topic needs an introduction (context). Please give context and logically connect the different paragraphs: right now they read like a collection of disconnected bulletpoints. Only traits are listed, not what they mean or their relevance for taxonomy.
What are the question to be answered? Utility of specific characters for taxonomy? Support for findings of this study? How does it all go together?
For example why is aedagus/sheet important? How does this new info relate to the Pygoluciola phylogeny? Can you diagnose Group 3 and 5 by aedagus/sheet? Make connection before you go to Luciolinae (which?) and how this relates to your study. I am sure this is clear for the expert minds of the authors, but not to the reader of Insects who may not be as immersed in the morphology of this group. Also please note: in the Discussion you usually start with your findings/details and expand back out to big picture (Literature), not the other way around.
Question: are you discussing the use of morphological traits that you identified as defining traits for the 5 Pygoluciola groups as potentially useful for other species in subfamily? And/or to make them more universally useful: how should specimens be treated, traits scored? (suggested improvements for next studies?)
1276: Luciolinae (other than/including Pygoluciola?)- does trait distinguish groups 3 and 5? – also shown for other Luciolinae? strong character for whole subfamily!?? -> future studies: traits need to be used? Be specific.
1279: “The significance of the aedeagal sheath in taxonomy was not realised until [53]”
Comment: please give at least year, if not name. Don’t force reader to scroll through references.
1279: aedagus/sheath section – this section reads defensive!
Why not reframe as: your study supports the significance of aedagus/sheet for Pygoluciola taxonomy. Difficulty with pulling aedagus/ -> what should taxonomist do in the future to fully get data? Aedeagal sheet often destroyed and therefore data not available for most specimen, including type species. What is your suggestion? What should the methods be for future studies?
1289: female morphology– [differentiation for what?], hooked bursa plates- what can you do with it/not?. Please be specific so reader can follow. Is this about limitations of morphological analysis? What is the point of this paragraph? Please clarify.
1294: terminal modification of male Luciolinae – what traits are these? start with your insights on Pygoluciola then discuss Luciolinae? Note: in Discussion you usually start with your findings/details and expand back out to big picture (Literature).
1301: context: first introduce copulation clamp and why absence/presence is key here for taxonomy (big picture) rather than just focusing on interpretation of specific structures. I am still not sure about this element after reading the manuscript.
Similarly, clarify the argument you are making with the information in paragraphs 1312, 1317, 1321.
Final comment: The morphological details (1276ff) should come earlier in the discussion. By ending with all the morphological details (and reasons why some traits are not useable, methods were not good, etc.) the reader is left with a negative impression and a bit lost (not sure what to do with all that information). This distracts from the contribution your study is making. Your discussion should end at the big picture/major insights. Discuss morphology first and what you learned from it about Pygoluciola relationships, which traits are key (and how future studies should collect/use them), then end with how morphological groups relate to the phylogeny and the phylogenetic relationships of Pygoluciola (integrate findings) and the other Luciolinae (literature) in this study. Which next study should be done (next question)? This way this manuscript ends on a high note with a clear take-home message.